# An astrocytic signaling loop for frequency-dependent control of dendritic integration and spatial learning

Kirsten Bohmbach [1], Nicola Masala[2], Eva M. Schönhense[1], Katharina Hill[1], André N. Haubrich[2], Andreas Zimmer [3], Thoralf Opitz[2], Heinz Beck [2,4] & Christian Henneberger [1,4,5] ✉

Dendrites of hippocampal CA1 pyramidal cells amplify clustered glutamatergic input by activation of voltage-gated sodium channels and N-methyl-D-aspartate receptors (NMDARs). NMDAR activity depends on the presence of NMDAR co-agonists such as D-serine, but how co-agonists influence dendritic integration is not well understood. Using combinations of whole-cell patch clamp, iontophoretic glutamate application, two-photon excitation fluorescence microscopy and glutamate uncaging in acute rat and mouse brain slices we found that exogenous D-serine reduced the threshold of dendritic spikes and increased their amplitude. Triggering an astrocytic mechanism controlling endogenous D-serine supply via endocannabinoid receptors (CBRs) also increased dendritic spiking. Unexpectedly, this pathway was activated by pyramidal cell activity primarily in the theta range, which required HCN channels and astrocytic CB1Rs. Therefore, astrocytes close a positive and frequency-dependent feedback loop between pyramidal cell activity and their integration of dendritic input. Its disruption in mice led to an impairment of spatial memory, which demonstrated its behavioral relevance.

Dendrites are the main neuronal input structure, receiving thousands of excitatory and inhibitory inputs. Their integration of incoming synaptic input is determined by passive and active mechanisms (for review see ref. [1]). Voltage dependent mechanisms can strongly amplify synchronous and spatially clustered synaptic glutamatergic inputs through generation of dendritic spikes. In dendrites of CA1 pyramidal cells of the hippocampus, this involves for instance voltage-dependent sodium channels and glutamate receptors of the N-methyl-D-aspartate subtype (NMDARs)[2–5]. Computationally, this represents, for example, a mechanism of coincidence detection[1] and numerous studies have described functional roles for dendritic spikes in CA1 pyramidal cells. For instance, dendritic spike generation plays a role for synaptic long-term plasticity in vitro[6] and dendritic spikes are strongly associated with somatic complex spike bursts in vivo[3]. The latter have been shown to occur in hippocampal place cells and to be spatially tuned in vivo[7,8]. More recently, studies using two-photon excitation imaging of dendritic Ca$^{2+}$ signals in behaving mice have directly visualized how dendritic Ca$^{2+}$ spikes can participate in the representation and formation of place fields[9,10].

The activation of NMDARs during dendritic spiking is driven by the binding of their ligand glutamate and postsynaptic depolarization. However, opening of NMDARs also requires the binding of a co-agonist, either glycine or D-serine[11,12]. At glutamatergic synapses of CA1 pyramidal cells, the level of saturation of the NMDAR co-agonist binding site with co-agonists can vary in an activity-dependent manner[13–17]. This indicated to us that the generation and properties

[1]Institute of Cellular Neurosciences, Medical Faculty, University of Bonn, Bonn, Germany. [2]Institute of Experimental Epileptology and Cognition Research, Medical Faculty, University of Bonn, Bonn, Germany. [3]Institute of Molecular Psychiatry, Medical Faculty, University of Bonn, Bonn, Germany. [4]German Center for Neurodegenerative Diseases (DZNE), Bonn, Germany. [5]Institute of Neurology, University College London, London, UK. ✉e-mail: christian.henneberger@uni-bonn.de

of dendritic spikes should be controlled by the availability of NMDAR co-agonists and its dynamic changes.

Among the two NMDAR co-agonists, the regulation of extracellular D-serine levels has recently attracted considerable attention[13–16,18,19]. We therefore asked if D-serine and documented mechanisms that control D-serine levels affect dendritic integration of CA1 pyramidal cells and spatial memory formation.

Here, we show that this is indeed the case and discover an unexpected frequency-dependent excitatory feedback loop between pyramidal cell activity and dendritic spiking that is mediated by NMDAR co-agonists and astrocytes. Importantly, disrupting this feedback loop at the level of astrocytes impairs spatial memory.

## Results

### Control of dendritic spiking by the NMDAR co-agonist D-serine

In a first set of experiments, we investigated dendritic spiking of CA1 pyramidal cells in acute hippocampal slices by iontophoretic glutamate application onto their dendrites in the stratum radiatum[20] (Fig. 1a, b) (all slice experiments throughout in the presence of picrotoxin). Increasing synaptic input was emulated by increasing the iontophoretic current and thus the amount of ejected glutamate. As expected, the amplitude of the somatically recorded voltage response increased (Fig. 1c, black traces and dots) until a threshold was passed and dendritic spikes composed of a fast and sodium channel-dependent component and a slow component[2,5] were detected (Fig. 1c, yellow dots and traces). Throughout this study, we used the

threshold stimulus that just evoked a dendritic spike and the amplitude of the slow component as parameters describing dendritic spiking. The first derivative of the membrane voltage (dV/dt) was used for orientation to clearly separate the two components (see Fig. 1c inset). We then used acute inhibition of NMDARs by APV to establish the extent to which NMDARs control dendritic spikes. APV increased the threshold stimulus and decreased the slow component of the dendritic spikes significantly (Fig. 1d) reflecting the well-established NMDAR-dependence of these dendritic spikes[2,4,5,21]. We next tested if increasing the occupancy of NMDAR co-agonist binding site by application of exogeneous D-serine (10 μM) alters dendritic spiking and found the opposite effect compared to NDMAR blockade (Fig. 1e). Note that in Fig. 1d, e the upper right panels illustrate traces recorded with the lowest stimulus eliciting a dendritic spike under baseline conditions (black trace before drug, colored trace in the presence of drug) and lower right panels illustrate traces recorded with the lowest stimulus eliciting a dendritic spike in the presence of the drug (black trace before drug, colored trace in the presence of drug). These results indicate that the NMDAR co-agonist binding site is not saturated in our experimental conditions and that its occupancy dynamically regulates dendritic spiking. Control experiments without drug application revealed that both stimulus threshold and the amplitude of the slow component were stable over time (not illustrated; baseline and after ~10 min; threshold: $0.56 \pm 0.08$ μA and $0.56 \pm 0.08$ μA, t(5) = 0.31, $n = 6$, $p = 0.76$, two-sided paired Student's $t$ test; slow component: $13.22 \pm 2.74$ mV and $13.39 \pm 2.69$ mV, $n = 6$, t(5) = 0.51, $p = 0.63$,

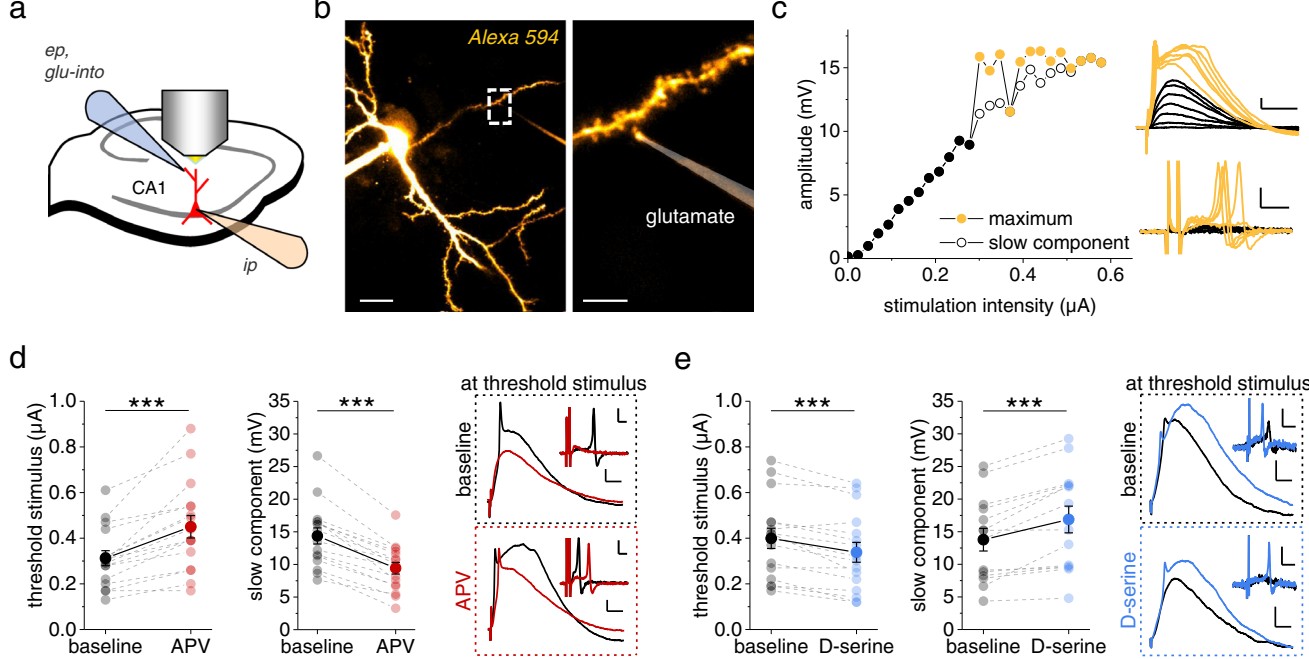

**Fig. 1 | Threshold and amplitude of dendritic spikes are controlled by N-methyl-D-aspartate receptor (NMDAR) co-agonist levels. a** Schematic of experiments. Whole-cell patch clamp recordings (ip, intracellular pipette, current clamp, CA1 pyramidal cells), iontophoretic glutamate application (ep, extracellular pipette), two-photon excitation (2PE) imaging of patched cell and iontophoresis pipette filled with Alexa Fluor 594 (40 μM and 50 μM, respectively). **b** Example of a typical recording (left panel: scale bar 20 μm; right panel: zoom in on dashed box in left panel, scale bar 5 μm). **c** Sample dependence of the somatic depolarizations on the iontophoretic current (left panel, filled circles: maximum amplitude, black below dendritic spike threshold and yellow above, empty circles: slow component amplitude). Right panels: corresponding sample traces. Upper right panel: scale bars 2 mV and 20 ms. Lower right panel: dV/dt, scale bars 5 mV/ms and 2 ms. In dV/dt the first two deflections represent the stimulus artifact. **d** NMDAR blockade (D-APV, 50 μM) increased the threshold stimulus (left panel, smallest iontophoretic

current eliciting a dendritic spike, $0.31 \pm 0.03$ μA vs. $0.45 \pm 0.05$ μA, $n = 16$, $p = 0.000077$) and reduced the slow component (middle panel, $14.36 \pm 1.23$ mV vs. $9.42 \pm 0.90$ mV, $n = 16$, $p = 0.00000022$). Right top panel: sample traces recorded with the baseline threshold stimulus. Right bottom panel: sample traces recorded with the threshold stimulus in D-APV (baseline: black traces; D-APV: red traces; scale bars 2 mV and 10 ms; insets: dV/dt, scale bars 5 mV/ms and 2 ms). **e** D-serine (10 μM) significantly decreased the threshold stimulus (left panel, $0.40 \pm 0.04$ μA vs. $0.34 \pm 0.04$ μA, $n = 16$, $p = 0.00011$) and increased the slow component (middle panel, $13.77 \pm 1.73$ mV vs. $16.87 \pm 2.05$ mV, $n = 14$, $p = 0.00024$). Right panels: sample traces of somatic voltage and dV/dt (insets) as before (upper traces with threshold stimulus during baseline; lower traces with threshold stimulus in D-serine, scale bars for voltage 2 mV and 10 ms and for insets 5 mV/ms and 2 ms). Two-sided paired Student's $t$ tests throughout. Data are expressed and displayed as mean ± s.e.m. Source data are provided as a Source Data file.

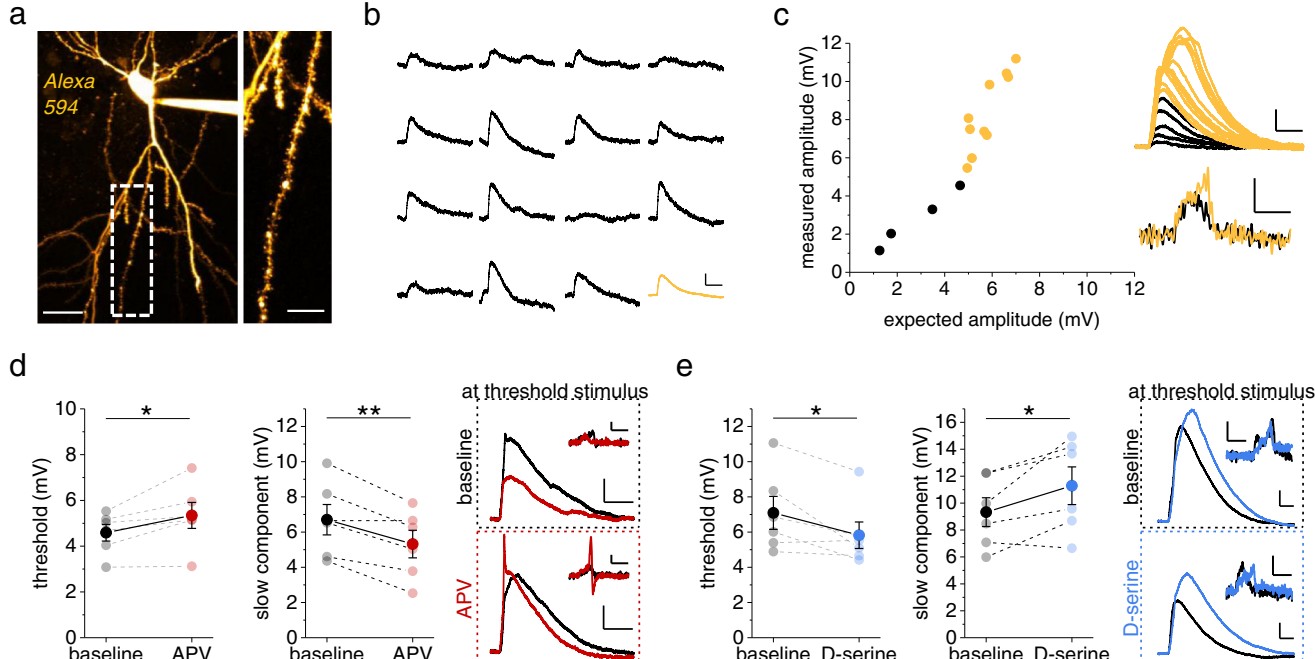

**Fig. 2 | D-serine controls threshold and amplitude of dendritic spikes evoked by glutamate uncaging. a** Two-photon glutamate uncaging. Example of a CA1 pyramidal cell filled with Alexa Fluor 594 (100 μM, left panel, scale bar 40 μm). Right panel: investigated dendrite (scale bar 10 μm). **b** Examples of uncaging-evoked somatic EPSPs (uEPSP, scale bars 0.5 mV and 20 ms, average uEPSP in yellow). **c** Left panel: comparison of the measured amplitudes of somatic responses evoked by quasi-simultaneous stimulation of a set of spines with the sum of single-spine uEPSPs (black: no spikes, yellow: spikes detected). Right top panel: individual responses corresponding to the left panel (scale bars 2 mV and 20 ms). Right bottom panel: dV/dt of traces above (scale bars 1 mV/ms and 5 ms). **d** Left panel: thresholds of dendritic spikes were significantly increased with blocked

NMDARs (D-APV, 40 μM, red, 4.59 ± 0.37 mV vs. 5.34 ± 0.57 mV, n = 6, p = 0.039). Middle panel: NMDAR inhibition decreased the slow component of the dendritic spike (6.70 ± 0.86 mV vs. 5.32 ± 0.78 mV, n = 6, p = 0.0097). Right panels: sample traces as in Fig. 1d (scale bars 2 mV and 50 ms, inset dV/dt scale bars 1 mV/ms and 10 ms). **e** Application of of D-serine (10 μM, blue) decreased the dendritic spike threshold (left panel, 7.09 ± 0.94 mV vs. 5.82 ± 0.75 mV, n = 6, p = 0.049) and increased its slow component (middle panel, 9.33 ± 1.07 mV vs. 11.29 ± 1.40 mV, n = 6, p = 0.045). Right panels: example traces as in Fig. 1e (scale bars 2 mV and 20 ms, inset dV/dt scale bars 1 mV/ms and 10 ms). Two-sided paired Student's t tests throughout. Data are expressed and displayed as mean ± s.e.m. Source data are provided as a Source Data file.

two-sided paired Student's t test). In addition, metabotropic glutamate receptors were not involved (Supplementary Fig. 1). Please note that here and throughout the study we used acute experimental manipulations whenever possible because absolute thresholds and amplitudes depend on the exact experimental configuration (e.g. iontophoretic pipettes, distance of stimulation site from cell body).

The bi-directional modification of dendritic spiking by NMDAR blockade and co-agonist site saturation was then confirmed using two-photon glutamate uncaging for stimulating an increasing number of dendritic spines[5,22] (Fig. 2a–c). Again, inhibition of NMDARs by APV decreased the amplitude of the slow spike component whereas exogenous D-serine had the opposite effect (Fig. 2d, e). For these uncaging experiments, the spike threshold was calculated as the expected excitatory postsynaptic potential (EPSP) amplitude at which dendritic spikes first occurred, i.e., as the arithmetic sum of the amplitudes of single spine uncaging-evoked EPSPs (uEPSPs) that was sufficient to trigger a dendritic spike. Similar to iontophoretic stimulation, NMDAR inhibition increased whereas D-serine application decreased the threshold (Fig. 2d, e). Note again that in Fig. 2d, e upper right panels illustrate traces recorded with the lowest stimulus eliciting a dendritic spike under baseline conditions (black trace before drug, colored trace in the presence of drug) and lower right panels illustrate traces recorded with the lowest stimulus eliciting a dendritic spike in the presence of the drug (black trace before drug, colored trace in the presence of drug). These observations were not due to changes of AMPAR-mediated synaptic transmission because neither iontophoretically evoked miniature-like EPSPs nor uEPSPs elicited by glutamate uncaging were affected by any of the drugs (p > 0.10 throughout, two-sided paired Student's t tests).

Our observations are in line with previous studies showing that NMDARs contribute to the late phase of dendritic spikes[2,5]. Similarly, blockade of NMDARs was previously shown to increase the dendritic spike threshold to a variable degree[2,5] and increasing NMDAR co-agonist levels decreased the spike threshold in our experiments. Therefore, our results indicate that boosting NMDAR function and thus NMDAR-mediated depolarization via co-agonist supply likely facilitates the activation of voltage-dependent ion channels, for example sodium channels mediating the initial fast component of dendritic spikes[2,5].

We and others have previously shown that astrocytic Ca²⁺ signaling can regulate the supply of D-serine to NMDARs[14,16,23,24]. Therefore, inducing astrocytic Ca²⁺ changes should affect dendritic spikes. A potent trigger of astrocytic Ca²⁺ signals is activation of their endocannabinoid receptors (CBRs)[16,23,25]. After confirming that activation of CBR signaling using WIN55 induced astrocytic Ca²⁺ signals (Fig. 3a, b), we tested if this manipulation also affects dendritic spiking. Indeed, we found that application of WIN55 lowered the stimulus threshold and increased the slow component of dendritic spikes (Fig. 3c). Importantly, we did not observe this when the experiments were performed in the presence of a blocker of the NMDAR co-agonist binding site (DCKA, 10 μM, Fig. 3d) and of exogenous D-serine to saturate the NMDAR co-agonist binding site (Fig. 3e). We also asked if indeed D-serine is the endogenous co-agonist involved. The identity of the relevant co-agonist can be established by incubating brain slices with D-amino acid oxidase (DAAO) to degrade endogenous D-serine[26-28]. Indeed, WIN55 did not affect dendritic spiking in the presence of DAAO (Fig. 3f). These findings indicate that activation of CBR induces astrocytic Ca²⁺

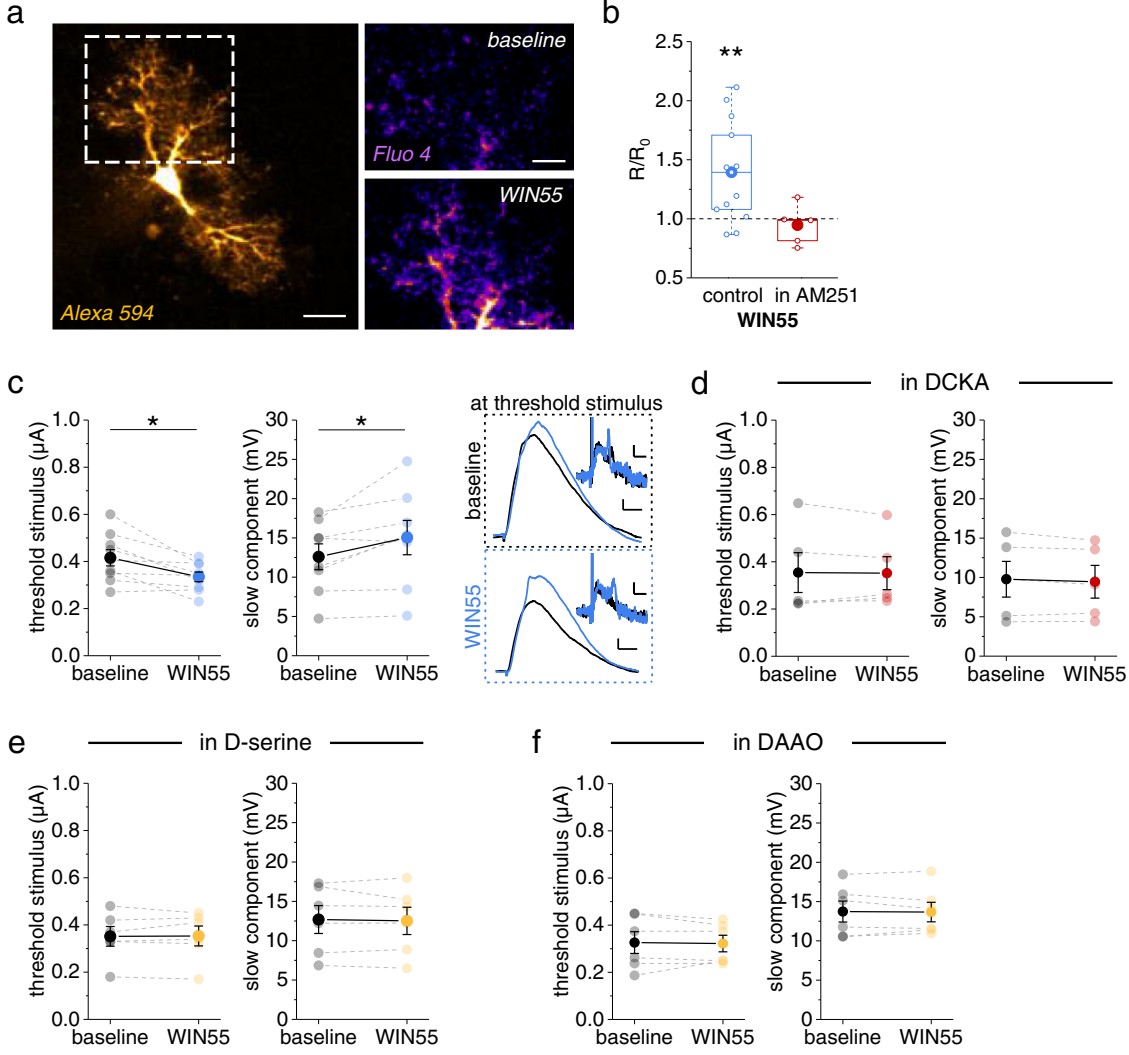

**Fig. 3 | Endocannabinoid receptor (CBR) activity controls astrocytic Ca²⁺ and dendritic spike threshold and amplitude. a** Example astrocyte filled with the Ca²⁺ indicator Fluo-4 (200-400 µM) and Alexa Fluor 594 (40 µM, left panel, scale bar 10 µm, whole-cell patch clamp). Right upper panel: baseline Fluo-4 fluorescence intensity. Right lower panel: Fluo-4 intensity during application of the CBR agonist WIN55 (WIN55,212-2, 10 µM). Both right panels: average of 10 frames, scale bar 5 µm. **b** Change of the fluorescence intensity ratio (R, Fluo-4 / Alexa 594) quantified as R in WIN55 relative to baseline (R₀, see methods section for panel **a, b**). Significant increase by WIN55 (blue, 1.39 ± 0.12, $n = 13$, $p = 0.005$, two-sided one-population Student's $t$ test) but not in the presence of the CBR inverse agonist AM251 (5 µM, red, 0.95 ± 0.08, $n = 5$, $p = 0.52$, two-sided one-population Student's $t$ test). **c** CBR activation by WIN55 (blue, 1 µM) decreased the threshold stimulus of dendritic spikes (left panel, 0.42 ± 0.03 µA vs. 0.33 ± 0.02 µA, $n = 9$, $p = 0.012$, two-sided paired Student's $t$ test) and increased their slow component amplitude (middle panel: 12.59 ± 1.65 mV vs. 15.04 ± 2.18 mV, $n = 8$, $p = 0.034$, two-sided paired Student's $t$ test). Right panels: sample EPSP traces (scale bars: 2 mV and 10 ms) and dV/dt (insets, scale bars: 1 mV/ms and 5 ms) at the baseline threshold stimulus

(upper traces) and at the threshold stimulus in WIN55 (lower traces). Both panels black for baseline and blue for WIN55. **d** The NMDAR co-agonist binding site antagonist DCKA (10 µM) occluded the WIN55 effect on the threshold stimulus (threshold stimulus: 0.35 ± 0.08 µA vs. 0.35 ± 0.07 µA, $n = 5$, $p = 0.88$, two-sided paired Student's $t$ test; slow component: 9.77 ± 2.27 mV vs. 9.45 ± 2.08 mV, $n = 5$, $p = 0.26$, two-sided paired Student's $t$ test). **e** In the presence of D-serine (10 µM), WIN55 fails to change dendritic spiking (threshold stimulus: 0.35 ± 0.04 µA vs. 0.35 ± 0.04 µA, $n = 6$, $p = 0.87$, two-sided paired Student's $t$ test; slow component: 12.69 ± 1.77 mV vs. 12.52 ± 1.73 mV, $n = 6$, $p = 0.64$, two-sided paired Student's $t$ test). **f** Presence of D-amino acid oxidase (DAAO, 0.17 U/ml) also prevented the WIN55 effect (threshold stimulus: 0.33 ± 0.05 µA vs. 0.32 ± 0.04 µA, $n = 6$, $p = 0.80$, two-sided paired Student's $t$ test; slow component: 13.73 ± 1.33 mV vs. 13.67 ± 1.24 mV, $n = 6$, $p = 0.84$, two-sided paired Student's $t$ test). Data are expressed and displayed as mean ± s.e.m. or in box plots. The box indicates the 25th and 75th, the whiskers the 5th and 95th percentiles, the horizontal line in the box the median and the mean is represented by a filled circle. Source data are provided as a Source Data file.

signals and modifies dendritic spiking, which involves the NMDAR co-agonist binding site and the co-agonist D-serine.

## Pyramidal cell activity controls dendritic spiking via the NMDAR co-agonist site

The observations above were obtained with pharmacological activation of astrocytic CBRs. We next explored if endogenous CBR activation leads to similar changes of dendritic spiking. Postsynaptic depolarization is a powerful trigger of endocannabinoid release[29,30]. We therefore induced postsynaptic depolarization of CA1 pyramidal

cells by antidromic stimulation of their axons in the alveus (Fig. 4a, Supplementary Fig. 2), which reliably activated ~40% of pyramidal cells over a broad range of stimulation frequencies (Supplementary Fig. 2e–j). We then investigated if this stimulation protocol induces astrocytic activity via endocannabinoid signaling. Indeed, antidromic activation of CA1 pyramidal cells evoked astrocytic Ca²⁺ signals, which were abolished by CBR inhibition using AM251 (Fig. 4a–c).

According to our previous results, such stimulation of pyramidal cell and astrocytic activity should increase the extracellular NMDAR co-agonist level and thereby affect dendritic spiking. We tested this

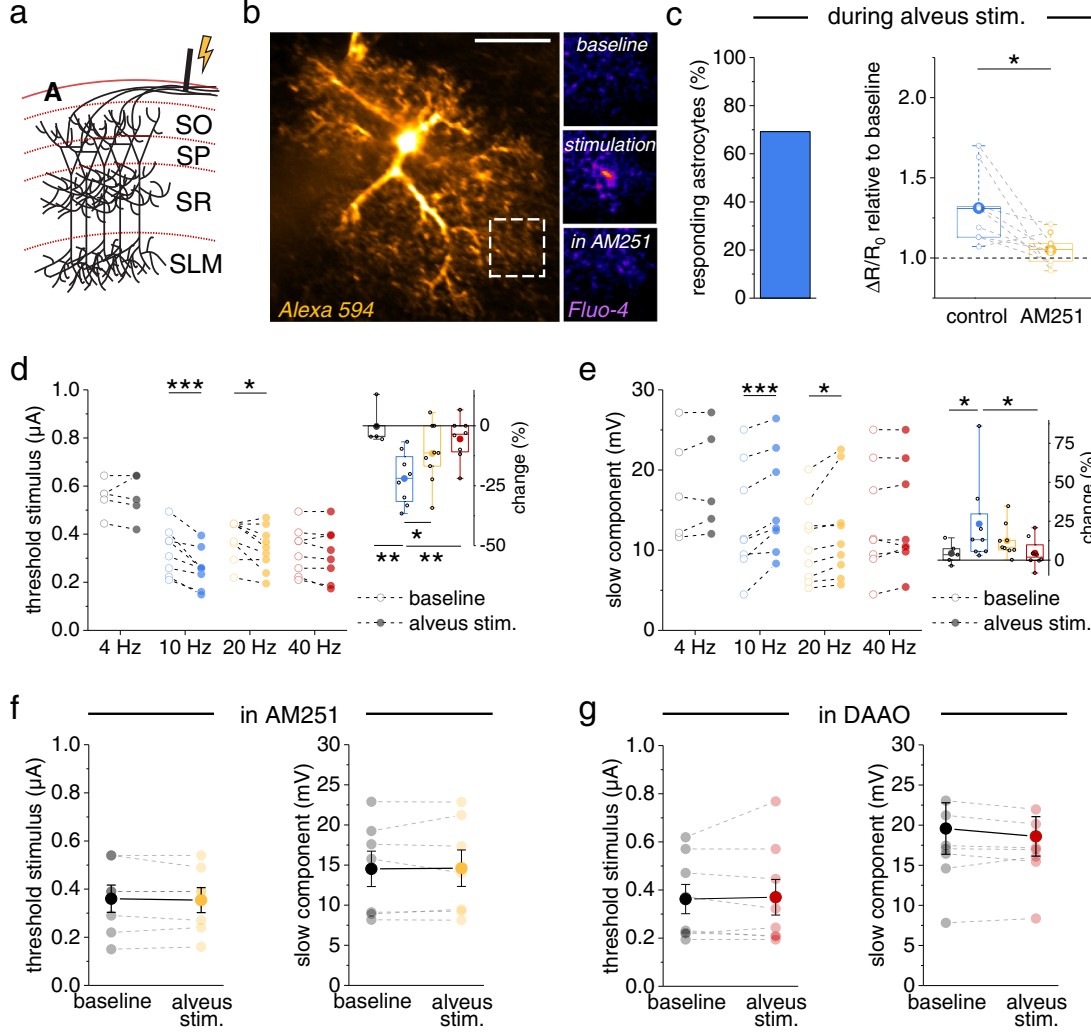

**Fig. 4 | Pyramidal cell activity increases astrocytic Ca²⁺ signaling and promotes dendritic spiking via CBRs and NMDAR co-agonist supply. a** Axonal simulation in the alveus (A) retrogradely activates CA1 pyramidal cells (SO, stratum oriens; SP, stratum pyramidale; SR, stratum radiatum; SLM, stratum lacunosum-molecular). **b** Example astrocyte (whole-cell patch clamp, 40 µM Alexa Fluor 594 and 400 µM Fluo-4, scale bar: 20 µm). Left panel: Alexa Fluor 594. Right panels, top to bottom: Fluo-4, average of 20 frames during baseline, alveus stimulation and alveus stimulation in the presence of the CBR inverse agonists AM251 (5 µM). Quantification of Ca²⁺ signals by the fluorescence intensity ratio (R, Fluo-4 / Alexa). Overall changes (ΔR) relative to resting value (R₀) during alveus stimulation compared to the baseline period before stimulation (see methods section for panels **b**, **c**). **c** 69.2% of the astrocytes showed an increase of at least 5% (left panel, 9 out of 13 cells). In the responders, the increase was blocked by AM251 (5 µM; 1.31 ± 0.07 vs 1.05 ± 0.03, n = 9, p = 0.016, two-sided paired Student's t test). **d** Left panel: the threshold stimulus of dendritic spikes was decreased by alveus stimulation at 10 Hz (0.33 ± 0.03 µA vs. 0.26 ± 0.03 µA, n = 8, p = 0.000071, repeated measures ANOVA and post-hoc Fisher's LSD) and 20 Hz (0.38 ± 0.03 µA vs. 0.34 ± 0.03 µA, n = 9, p = 0.031, Friedman test and post-hoc two-sided Wilcoxon signed-rank test) but not 4 Hz (0.55 ± 0.03 µA vs. 0.55 ± 0.04 µA, n = 5, p = 1, two-sided paired Student's t test, separate set of experiments) and 40 Hz (0.33 ± 0.03 µA vs. 0.32 ± 0.04 µA, n = 8, p = 0.31, repeated measures ANOVA and post-hoc Fisher's LSD). Open circles: baseline. Filled circles: alveus stimulation. Right panel: change relative to baseline

(one-way ANOVA with post-hoc Fisher's LSD). **e** Left panel: corresponding changes of the slow component (10 Hz: 13.70 ± 2.47 mV vs. 15.74 ± 2.29 mV, n = 8, p = 0.000051, repeated measures ANOVA with post-hoc Fisher's LSD; 20 Hz: 9.75 ± 1.37 mV vs. 11.08 ± 1.81 mV, n = 8, p = 0.022, repeated measures ANOVA with post-hoc Fisher's LSD; 4 Hz: 17.97 ± 2.98 mV vs. 18.61 ± 2.94 mV, n = 5, p = 0.23, two-sided paired Student's t test, separate set of experiments; 40 Hz: 13.70 ± 2.47 mV vs. 14.00 ± 2.39 mV, n = 8, p = 0.41, repeated measures ANOVA with post-hoc Fisher's LSD). Open circles: baseline. Filled circles: alveus stimulation. Right panel: change relative to baseline (Kruskal-Wallis test with post-hoc two-sided Mann–Whitney U-tests). **f** In the presence of AM251 (5 µM, yellow), no effect of alveus stimulation (20 Hz) on dendritic spikes (threshold stimulus: 0.36 ± 0.06 µA vs. 0.35 ± 0.05 µA, n = 7, p = 0.54, two-sided paired Student's t test; slow component amplitude: 14.53 ± 2.21 mV vs. 14.61 ± 2.27 mV, n = 7, p = 0.85, two-sided paired Student's t test). **g** Presence of D-amino acid oxidase (DAOO, 0.17 U/ml, red) also prevent the effect of alveus stimulation (10 Hz) (threshold stimulus: 0.36 ± 0.06 µA vs. 0.37 ± 0.07 µA, n = 8, p = 0.73, two-sided paired Student's t test; slow component: 19.58 ± 3.21 mV vs. 18.60 ± 2.46 mV, n = 8, p = 0.27, two-sided paired Student's t test). Data are expressed and displayed as mean ± s.e.m. or in box plots. The box indicates the 25th and 75th, the whiskers the 5th and 95th percentiles, the horizontal line in the box the median and the mean is represented by a filled circle. Supplementary Table 2 for further information. Throughout panels *p < 0.05, **p < 0.01, ***p < 0.001. Source data are provided as a Source Data file.

prediction by monitoring dendritic integration before and during alveus stimulation at various frequencies and found that stimulation at 10 and 20 Hz but not at 4 and 40 Hz reduced the stimulus threshold of dendritic spikes (Fig. 4d) and increased the slow component of dendritic spikes (Fig. 4e). This reveals that population activity of pyramidal cells facilitates the generation of dendritic spikes in a frequency-

dependent manner. This is not because of a modulation of AMPAR-mediated responses because alveus stimulation did not change AMPAR-mediated iontophoretically evoked miniature-like EPSPs (p > 0.40 for 4 Hz, 10 Hz, 20 Hz and 40 Hz, two-sided paired Student's t tests). Similar experiments in the presence of the CBR antagonist AM251 failed to change dendritic spiking and thus demonstrate that

CBRs mediate this positive, frequency-dependent feedback loop (alveus stimulation at 20 Hz, Fig. 4f). A change of dendritic spiking was also not observed when we performed this experiment in the presence of D-serine (not illustrated; alveus stimulation at 20 Hz, threshold: $0.44 \pm 0.05\,\mu A$ and $0.45 \pm 0.05\,\mu A$, $n = 6$, $t(5) = 1$, $p = 0.36$, two-sided paired Student's $t$ test; slow component: $20.34 \pm 2.54\,mV$ and $20.05 \pm 2.46\,mV$, $n = 6$, $t(5) = 0.55$, $p = 0.61$, two-sided paired Student's $t$ test). Also, repeating our experiments in the presence of DAAO prevented a modulation of dendritic spiking by alveus stimulation (10 Hz), which reveals that D-serine is involved (Fig. 4g). Consistent with a transient release of D-serine triggered by pyramidal cell activity, dendritic spiking returned to its baseline within five minutes after alveus stimulation (Supplementary Fig. 3).

## Mechanism underlying the frequency dependence of dendritic spiking modulation

Next, we dissected the cellular mechanisms underlying the unexpected frequency dependence of activity-induced changes of dendritic integration. While it may not be surprising that low frequency stimulation is insufficient to activate this pathway, the absence of changes of dendritic integration at 40 Hz was less expected. We had already established that pyramidal cells activated by alveus stimulation reliably follow stimulation frequencies between 4 and 40 Hz (Supplementary Fig. 2h–j). Thus, the frequency dependence could be caused at the level of CBR-dependent astrocyte $Ca^{2+}$ signaling or co-agonist release, or at the level of excitability of pyramidal cell dendrites, the presumed location of endocannabinoid release. If the latter is the case, then astrocytic $Ca^{2+}$ signals are expected to also display a frequency-dependent modulation by alveus stimulation. Indeed, alveus stimulation at 10 Hz was more efficiently inducing astrocytic $Ca^{2+}$ transients than at 40 Hz (Fig. 5a–c, Supplementary Fig. 4), which again could arise from frequency-dependent astrocytic $Ca^{2+}$ signaling or pyramidal cell endocannabinoid release. Because a frequency-dependence of pyramidal cell excitability can be conferred by HCN channels to pyramidal cells and especially to their dendrites[31,32], we next tested their involvement by performing experiments in the presence of the HCN inhibitor ZD7288. In these experiments, we first established the whole-cell patch clamp configuration, washed in ZD7288 and compensated the hyperpolarization of the recorded pyramidal cell due to HCN blockade ($-4.99 \pm 1.06\,mV$, $n = 18$) by a constant current injection before testing dendritic spiking. Indeed, we found that alveus stimulation at 10 Hz was not affecting the stimulus threshold and slow components of dendritic spikes in the presence of ZD7288 (Fig. 5d, e, yellow). However, direct activation of CBRs by WIN55 was still effective (Fig. 5d, e, blue) indicating that the frequency dependence is likely caused by dendritic mechanisms that involve HCN channels. In line with this, ZD7288 also occluded the increase of astrocytic $Ca^{2+}$ events triggered by alveus stimulation at 10 Hz (Fig. 5f, g, Supplementary Fig. 5).

## Astrocytic type 1 endocannabinoid receptors (CB1Rs) in dendritic spiking and place memory

Our results indicate that astrocytic CBRs close a positive feedback loop between pyramidal cell activity and their dendritic spiking via NMDAR co-agonist signaling. To directly establish astrocytic CBR involvement, we deleted astrocytic CB1Rs by injecting a transgenic mouse line GLASTcreERT2[33] crossed with the CB1R fl/fl line[34] and the reporter line (flox-stop-tdTomato)[35] with tamoxifen (aCB1KO). We used sham injected littermates (sham) and wildtype animals (WT) as controls (Fig. 6). We first confirmed that tdTomato expression was near-exclusively restricted to astrocytes in the hippocampal CA1 region (Supplementary Fig. 6). In addition, no major differences of basic properties of CA3-CA1 synaptic transmission were detected between slices from these mice (Supplementary Fig. 7, Supplementary Table 1). If astrocytic CB1Rs mediate frequency-dependent changes of dendritic spiking, then their removal in aCB1KO mice should disrupt this positive

feedback. We therefore tested dendritic spiking evoked by iontophoretic glutamate application in the territory of a tdTomato-expressing astrocyte before and during alveus stimulation (Fig. 6a, b). Indeed, we found that alveus stimulation at 10 Hz was unable to affect the threshold stimulus and the amplitude of the slow component of dendritic spikes, while this effect was, however, preserved in slices from WT and sham animals (Fig. 6c, d). Importantly, application of exogenous D-serine still modified dendritic spiking in aCB1KO mice (red data points, Fig. 6c, d), as observed before (Fig. 1), indicating that astrocytic CB1Rs are upstream of NMDAR co-agonist supply.

This shows that astrocytic CB1Rs mediate the positive and frequency-dependent feedback loop between pyramidal cell activity and their dendritic integration (see Fig. 7 for a schematic illustration). Our experiments also identify astrocytic CB1Rs as the link between the activity dependent release of endocannabinoids from CA1 pyramidal cells and the astrocytic control of D-serine supply and its effect on dendritic spiking.

Because NMDAR-driven dendritic spikes in CA1 pyramidal cells have been implicated in the formation of place memory (for review see ref. [36]), we next asked if genetic deletion of astrocytic CB1Rs impairs behaviors that require the encoding of a location. First, we tested object location memory by allowing animals to explore two objects in a first session (acquisition) and then moving one of the objects to a new location (test, Fig. 8a). An increased exploration of the displaced object during the recall trial (test) indicates that the animal has encoded the initial location and is able to discriminate the displaced object. Sham injected and aCB1KO animals showed neither an initial preference for object locations during acquisition ($0.08 \pm 0.03$ vs. $-0.002 \pm 0.07$, $n = 10/12$, $t(20) = 0.98$, $p = 0.34$, two-sided Student's $t$ test), nor did the overall exploration of the objects differ on the test day (test, Fig. 8b). We also did not detect differences between the groups regarding the exploration of the arena (Supplementary Fig. 8a–d) and the spontaneous alternation in the Y-maze, indicating intact spatial working memory (Supplementary Fig. 8e–h). In contrast, aCB1KO mice did not display a preference for the displaced object in the test trial compared to sham injected animals (Fig. 8c), indicating impaired object location memory.

We next examined how well sham injected and aCB1KO animals adapt to changes in the location of an aversive stimulus. We used a place avoidance test, in which mice first learn to avoid the location of an automatically triggered air puff at a specific position (Fig. 9a, position A) in an O-shaped maze for two days (day 1 and 2, probe trial on day 3). Subsequently, the aversive stimulus is switched to the opposite side of the O-maze (Fig. 9a, position B), requiring mice to learn the new aversive stimulus location. The spatial component in such tests is believed to be processed by the hippocampus whereas the integration of the aversive component requires the amygdala[37,38]. Initially, both groups of animals rapidly learned to avoid the air puff (day 1 and 2) and reacted similarly to the removal of the air puff from position A on day 3 (Fig. 9b, Supplementary Fig. 8i). However, when the air puff was moved to position B on day 4, aCB1KO mice more often triggered the air puff in the beginning of the experiments (Fig. 9c) and avoided position B less than their control littermates (Fig. 9d, Supplementary Fig. 8i–l). Together these results indicate that aCB1KO animals do learn to avoid an aversive stimulus as efficiently as controls but display a transient deficit (Fig. 9c, right panel) when the spatial context needs to be updated.

## Intact hippocampal astrocytic $Ca^{2+}$ signaling is required for the frequency dependent modulation of dendritic integration and object location memory

Our experiments predict that suppressing astrocytic $Ca^{2+}$ signaling should weaken the frequency dependent modulation of dendritic spiking, because astrocytic $Ca^{2+}$ signaling is downstream of astrocytic

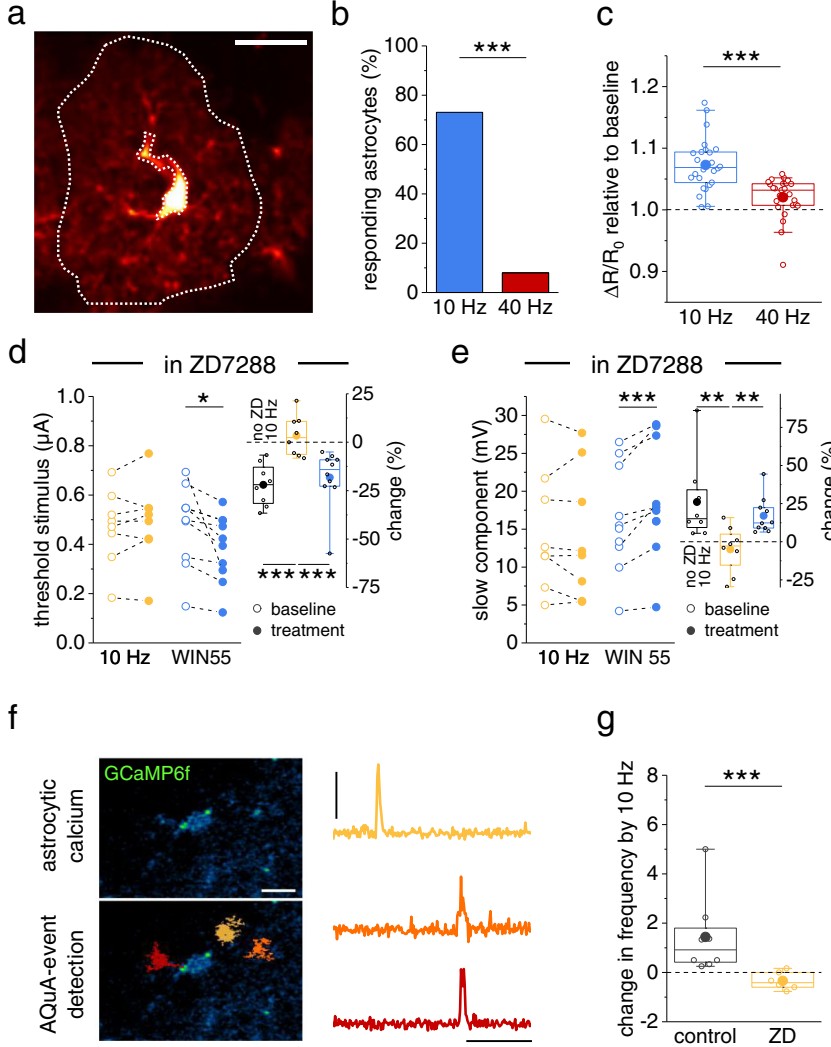

**Fig. 5 | Mechanisms underlying the frequency-dependence of dendritic spike modulation. a** Astrocyte expressing GCaMP5g (Supplementary Fig. 4) and tdTomato (red, scale bar: 20 μm; region of interest: dotted while line, sparing the cell body and major branches). Quantification of Ca²⁺ signals by the fluorescence intensity ratio (R, GCaMP5g / tdTomato) and its changes (ΔR) over its resting value ($R_0$). Comparison of $\Delta R/R0$ during alveus stimulation with the baseline period before stimulation (see methods section for panels **a–c**). **b** Percentage of astrocytes with Ca²⁺ signaling increase of >5% was significantly larger with 10 Hz (blue) stimulation than with 40 Hz (73.1 % vs. 9.52 %, two-sided Fisher's exact test, $p = 0.0000022$, $n = 19$ out of 26 cells and 2 out of 25 cells). **c** Change of $\Delta R/R0$ during alveus stimulation across all cells (10 Hz vs 40 Hz: 1.07 ± 0.008 vs. 1.02 ± 0.007, $n = 26$ and 25, $p = 0.0000046$, two-sided Mann–Whitney U-test). **d, e** ZD7288 (10 μM) prevented the modulation of dendritic spiking by alveus stimulation while the effect of direct CBR stimulation by WIN55 (1 μM) was preserved. **d** Threshold stimulus. Left panel: 10 Hz (0.47 ± 0.05 μA vs. 0.49 ± 0.06 μA, $n = 8$, $p = 0.33$, two-sided paired Student's t test) and WIN55 (0.48 ± 0.05 μA vs. 0.38 ± 0.04 μA, $n = 10$, $p = 0.025$, two-sided paired Student's t test). Right panel: change during treatment (solid circles, left panel) relative to baseline (open circles, left panel), Kruskal-Wallis test with post-hoc two-sided Mann–Whitney U-test.

**e** Slow component. Left panel: 10 Hz (14.75 ± 2.87 mV vs. 14.27 ± 3.05 mV, $n = 8$, $p = 0.52$, two-sided paired Student's t test) and WIN55 (16.24 ± 2.21 mV vs. 18.79 ± 2.41 mV, $n = 10$, $p = 0.0010$, two-sided paired Student's t test). Right panel: change during treatment (solid circles, left panel) relative to baseline (open circles, left panel), Kruskal-Wallis test with post-hoc two-sided Mann–Whitney U-test. Right panels of (**d** and **e**). The white control bar with 10 Hz alveus stimulation represents control data from Fig. 4d. **f** Ca²⁺ events in astrocytes expressing GCaMP6f (**f**, upper left panel, scale bar 10 μm) detected using AQuA[73] (lower left panel: sample events in orange, yellow and red; right panel: corresponding traces, scale bars: 0.5 ΔF/F and 1 min). **g** Change in Ca²⁺ event frequency during alveus stimulation ($f_{10Hz}$ / $f_{baseline}$ − 1) across all cells. Control (gray) vs. ZD7288 (yellow, ZD, 10 μM): 1.44 ± 0.56 vs. −0.34 ± 0.15 ($p = 0.00067$, $n = 8$ and 6 cells from 4 and 3 mice, two-sided Mann–Whitney U-test). Two-sided one sample Wilcoxon Signed Rank tests: $p = 0.008$ for control and $p = 0.13$ for ZD ("Method" section for panels **f, g**). Data are expressed as mean ± s.e.m. in the legend and displayed in box plots. The box indicates the 25th and 75th, the whiskers the 5th and 95th percentiles, the horizontal line in the box the median and the mean is represented by a filled circle. Supplementary Table 2 for further information. Throughout panels *$p < 0.05$, **$p < 0.01$, ***$p < 0.001$. Source data are provided as a Source Data file.

CB1R activity and involved in the astrocytic control of extracellular D-serine levels (see above). To test this prediction, we used adeno-associated viruses to express a human Ca²⁺ pump bilaterally (hPMCA2w/b), which was previously shown to strongly attenuate astrocytic Ca²⁺ signaling in the hippocampus and striatum (CalEx)[39]. We first confirmed that this technique specifically targets local astrocytes in wildtype mice (Fig. 10a, Supplementary Fig. 9). Next, we tested if alveus stimulation promotes dendritic spiking in acute hippocampal

slices with astrocytes expressing hPMCA2w/b, which was not the case (CalEx, Fig. 10b, c). In contrast, the activity-dependent change of dendritic integration was intact in control experiments in acute hippocampal slices with viral expression of tdTomato in astrocytes (sham, Fig. 10b, c). This demonstrates that astrocytic expression of the Ca²⁺ pump hPMCA2w/b indeed disrupts the activity-dependent positive feedback loop between pyramidal cell activity and their dendritic spiking.

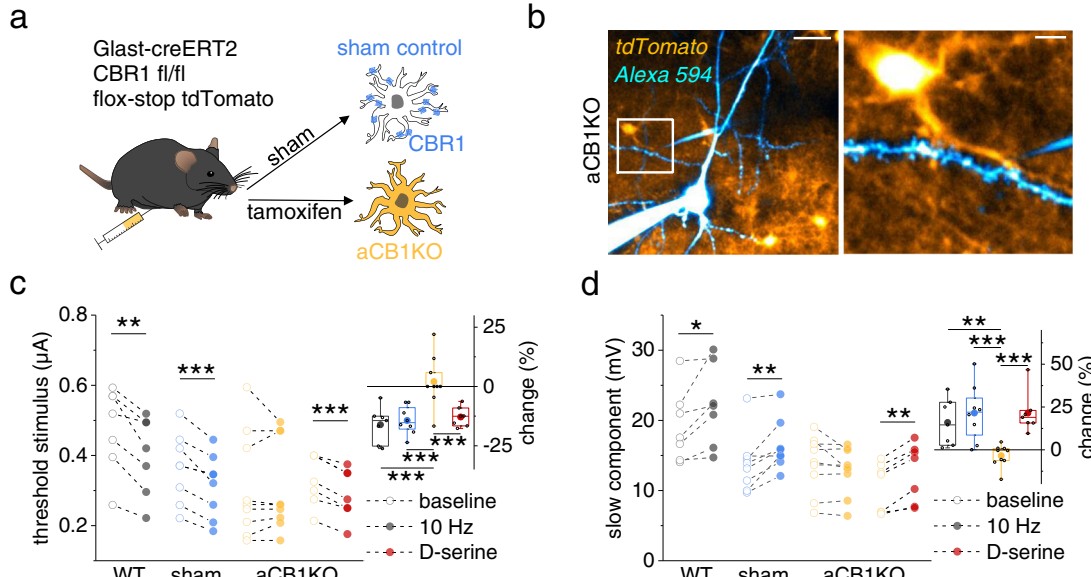

**Fig. 6 | Deletion of astrocytic type 1 CBRs (CB1Rs) disrupts the frequency-dependent modulation of dendritic spiking by pyramidal cell activity.**
**a** Generation of sham and aCB1KO mice. **b** Sample CA1 pyramidal cell filled with Alexa Fluor 594 (40 μM, blue, whole-cell patch clamp), an iontophoresis pipette (blue) and expression of tdTomato (indicator of astrocytic CB1R deletion, aCB1KO). Left panel: scale bar 20 μm. Right panel: enlarged white box from right panel, scale bar 5 μm. **c** Dendritic spike threshold decrease during 10 Hz alveus stimulation absent aCB1KO mice but preserved in wildtype (WT) and sham control mice (left panel; WT: $0.48 \pm 0.05$ μA vs. $0.40 \pm 0.04$ μA, $n = 7$, $p = 0.0026$, two-sided paired Student's $t$ test; sham: $0.36 \pm 0.03$ μA vs. $0.31 \pm 0.03$ μA, $n = 8$, $p = 0.00071$, two-sided paired Student's $t$ test; aCB1RKO: $0.31 \pm 0.05$ μA vs. $0.31 \pm 0.04$ μA, $n = 9$, $p = 1.00$, two-sided Wilcoxon signed-rank test). Decrease after D-serine application (10 μM) is preserved in aCB1KO mice ($0.33 \pm 0.03$ μA vs. $0.29 \pm 0.03$ μA, $n = 7$, $p = 0.000096$, two-sided paired Student's $t$ test). Right panel: change (filled circles,

left panel) relative to baseline (open circles, left panel). One-way ANOVA with post-hoc Fisher's LSD. **d** Corresponding slow component changes during alveus stimulation (WT: $19.18 \pm 1.94$ mV vs. $22.14 \pm 2.19$ mV, $n = 7$, $p = 0.031$, two-sided paired Student's $t$ test; sham: $13.83 \pm 1.50$ mV vs. $16.41 \pm 1.29$ mV, $n = 8$, $p = 0.0056$, two-sided paired Student's $t$ test; aCB1KO: $13.73 \pm 1.36$ mV vs. $12.83 \pm 1.14$ mV, $n = 9$, $p = 0.14$, two-sided paired Student's $t$ test) and D-serine application (aCB1KO: $10.51 \pm 1.33$ mV vs. $12.70 \pm 1.56$ mV, $n = 7$, $p = 0.0011$, two-sided paired Student's $t$ test). Right panel: change (filled circles, left panel) relative to baseline (open circles, left panel). One-way ANOVA with post-hoc Fisher's LSD. Data are expressed as mean ± s.e.m. in the legend and displayed in box plots. The box indicates the 25th and 75th, the whiskers the 5th and 95th percentiles, the horizontal line in the box the median and the mean is represented by a filled circle. Throughout panels *$p < 0.05$, **$p < 0.01$, ***$p < 0.001$. Supplementary Table 2 for further information. Source data are provided as a Source Data file.

We next tested if the same manipulation also affects object location memory. Sham injected and CalEx mice showed no differences in the open field test (Supplementary Fig. 10), in their preference for object locations during acquisition ($0.03 \pm 0.10$ vs. $-0.08 \pm 0.08$, $n = 7$ and 5, $t(10) = 0.85$, $p = 0.41$, two-sided Student's $t$ test), or in the overall exploration times of object during the test trial (Fig. 10e). However, CalEx mice were unable to identify the displaced object as they showed, in contrast to the sham-injected mice, no preferential exploration of the moved object (Fig. 10f).

Therefore, astrocytic expression of the Ca²⁺ pump hPMCA2w/b disrupts both the activity-dependent modulation of dendritic spiking and object location memory, which is identical to what was found with astrocytic CB1 receptor deletion and thus further strengthens the causal link between the two observations. Although viral expression could not be strictly limited to the hippocampus, we primarily targeted hippocampal astrocytes for hPMCA2w/b expression (Fig. 10a, Supplementary Fig. 9). We found that this is sufficient to perturb object location memory, which is in line with previous studies implicating the hippocampus in object location memory[40,41].

## Discussion

A main hypothesis of our study was that dendritic spiking of CA1 pyramidal cells is controlled by the extracellular level of NMDAR co-agonists. Indeed, we found that exogenous D-serine and other manipulations of co-agonist levels transiently affected dendritic spiking in CA1 stratum radiatum so that an increase of extracellular co-agonist levels reduced the dendritic spike threshold and increased their amplitude, and vice versa. This indicates that the occupancy of

the NMDAR co-agonist binding site dynamically determines NMDAR function and thereby dendritic integration and spiking.

Our study also reveals that the activity of pyramidal cell populations controls dendritic spiking in a frequency-dependent manner via NMDAR co-agonist supply, which depends on astrocytic CB1Rs (see Fig. 7 for a schematic). Therefore, astrocytes close a positive feedback loop between pyramidal cell activity and their own dendritic spiking, which displays a bell-shaped frequency dependence with a peak at ~10 Hz. While pyramidal cell activity at 4 Hz did not affect dendritic spiking, stimulation at 10 Hz decreased the dendritic spike threshold and increased the dendritic spike amplitude. A straightforward explanation is that the dendrites of CA1 pyramidal cells are increasingly depolarized as the stimulation frequency increases and more efficiently activate NMDAR co-agonist release via CB1R-dependent astrocytic Ca²⁺ signaling, because the latter was shown to increase as the duration of direct neuronal depolarization is prolonged[25]. This raises the question why the effect on dendritic spiking decreases again at a stimulation frequency of 20 Hz and disappears at 40 Hz. Having excluded that the somatic firing of pyramidal cells cannot follow a 40 Hz stimulation, another possibility is that the frequency dependence is of dendritic origin. To test that hypothesis, we inhibited HCN-mediated currents, which are particularly strong in dendrites of CA1 pyramidal cells and confer a frequency-dependence to their excitability[32,42]. This prevented the modulation of dendritic spiking and of astrocytic Ca²⁺ signals by pyramidal cell activity at 10 Hz and suggests that indeed dendritic excitability controls the modulation of dendritic spiking, because HCN expression by other neuronal subtypes is not relevant in our experimental conditions. For instance, inhibition

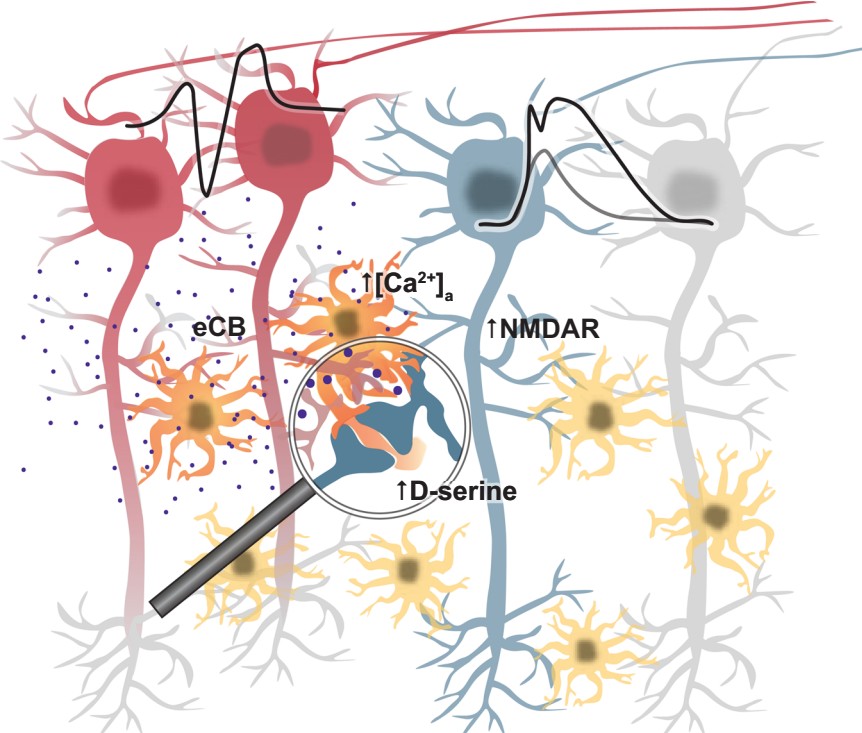

**Fig. 7 | Schematic of the astrocytic mechanism closing a positive feedback loop targeting dendritic spiking.** Active pyramidal cells (red) trigger the release of endocannabinoids, which act on astrocytic CB1 receptors and thereby increase extracellular D-serine levels. This leads to increased opening of NMDARs, a lower threshold of dendritic spikes and a higher NMDAR component. This positive feedback loop is primarily engaged by theta-like activity of pyramidal cells.

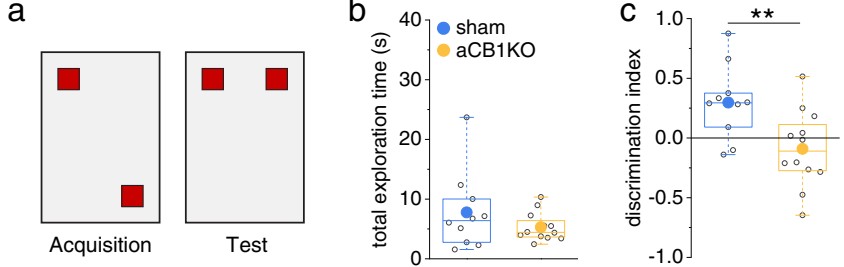

**Fig. 8 | Disruption of astrocytic CB1R-mediated signaling impairs formation of spatial memory. a** Schematic arena for testing object location memory. Two identical objects (red squares) are explored by the animals (10 min) during acquisition. During testing, one object is moved to a new location and mice are allowed for freely explore the objects (5 min). **b** Total exploration time did not differ between sham control animals and animals with conditional deletion of CB1Rs in astrocytes (aCB1KO) (7.78 ± 2.07 s vs. 5.30 ± 0.69 s, $n = 10$ and 12 animals, $p = 0.54$, two-sided Mann–Whitney U-test). **c** Sham control mice explored the object in the novel location significantly more whereas aCB1KO mice did not discriminate (0.30 ± 0.10 vs. −0.09 ± 0.09, $n = 10$ and 12 animals, $p = 0.0098$, two-sided Student's $t$ test). Data are displayed in box plots, in which the box indicates the 25th and 75th, the whiskers the 5th and 95th percentiles, the horizontal line in the box the median and the mean is represented by a filled circle. Source data are provided as a Source Data file.

of HCN could hyperpolarize the dendritic trees of the stimulated pyramidal cell population thereby interfering with endocannabinoid release or inhibit dendritic resonance[32]. In addition, the declining modulation of dendritic integration at higher frequencies could reflect a reduced propagation of somatic action potentials into the dendritic trees of stimulated cells because of increased dendritic inactivation of voltage-dependent sodium channels[22]. Although the mechanisms underlying the frequency dependence clearly requires further dissection, our results demonstrate that extracellular NMDAR co-agonist levels are controlled by pyramidal cell activity via astrocytic CB1Rs with a bell-shaped frequency dependence peaking around the higher end of the theta frequency range.

Such a feedback loop could be relevant for the formation of spatial memories for several reasons. First, the investigated dendritic spiking has been implicated in the encoding of spatial information (see

"Introduction"). Second, NMDARs participate in the generation of dendritic plateau potentials, which are driven by synaptic input from the entorhinal cortex and through CA3-CA1 synapses and contribute to place field formation[43,44]. Third, the observed peak of the dendritic spike modulation is in the theta range, which is an activity pattern that occurs for instance during spatial exploration[45]. Indeed, animals without astrocytic CB1Rs or with attenuated astrocytic Ca²⁺ signaling showed an impaired performance in an object location memory test. This reveals that astrocytic CB1Rs and NMDAR co-agonist signaling are important for encoding and/or storing object locations, in addition to their role in object recognition memory[16]. Interestingly, the deliberate activation of astrocytes has recently been shown to play a role in contextual and object recognition memory too: Chemogenetic and optogenetic activation of hippocampal astrocytes improves the acquisition of contextual memory[13] and optogenetic activation of

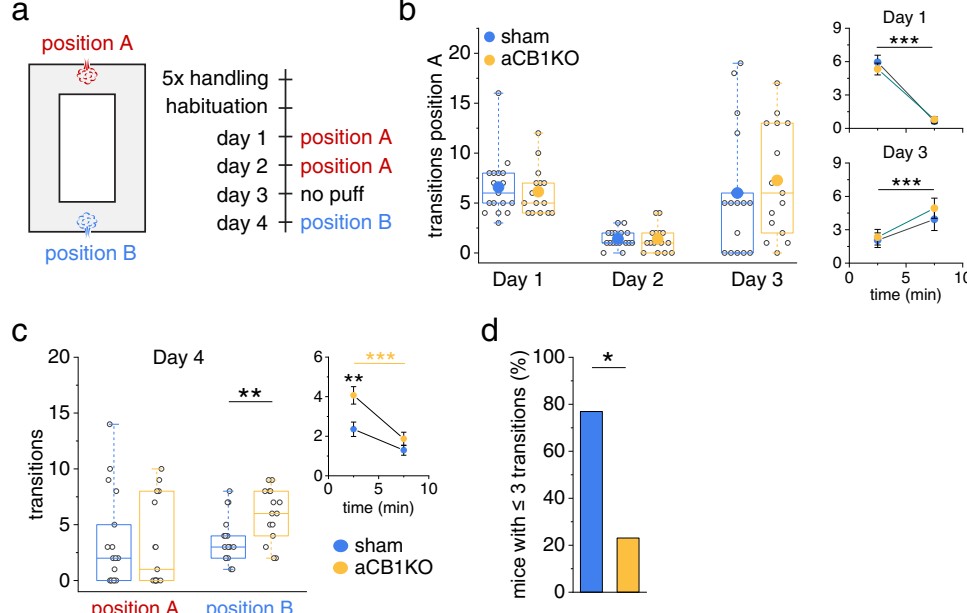

**Fig. 9 | Disruption of astrocytic CB1R-mediated signaling leads to a transient deficit in updating spatial memory. a** Passive place avoidance test with two possible positions for a triggered air puff (each session 10 min). **b** Similar air puff activation during acquisition (day 1) in sham and aCB1KO mice (6.59 ± 0.73 vs. 6.13 ± 0.62, $n = 17$ and 15 animals, $p = 0.62$, two-sided Mann–Whitney U-test). Activations binned into five-minute time intervals (small insets) decrease over time ($n = 17$ and 15 animals, $p < 0.00001$, two-way repeated measures ANOVA) without a difference between groups ($n = 17$ and 15 animals, $p = 0.64$, two-way repeated measures ANOVA). Day 2: no difference in the total number of activations (1.41 ± 0.21 vs. 1.40 ± 0.34, $n = 17$ and 15 animals, $p = 0.73$, two-sided Mann–Whitney U-test). Removal of air puff from position A (day 3): number of transitions at position A increased over time for both groups (small inset, sham control: 2.06 ± 0.66 vs. 3.94 ± 1.01, aCB1KO: 2.33 ± 0.70 vs. 4.93 ± 0.91, $n = 17$ and 15 animals, time $p = 0.00031$, treatment $p = 0.55$, time × treatment $p = 0.52$, two-way repeated measures ANOVA); no significant difference between groups (6.00 ± 1.50 vs. 7.27 ± 1.45, $n = 17$ and 15 animals, $p = 0.45$, two-sided Mann–Whitney U-test). **c** Left panel: on day 4 (reversal), the air puff was moved to position B. Animals passed position A without a difference between groups (transitions, 3.47 ± 1.04 vs. 3.27 ± 1.02, $n = 17$ and 15 animals, $p = 0.77$, two-sided Mann–Whitney U-test).

aCB1KO mice activated the air puff at position B more frequently than sham control mice (3.65 ± 0.50 vs. 5.93 ± 0.61, $n = 17$ and 15, $p = 0.0066$, two-sided Student's $t$ test). Right panel: number of air puff activations binned in five minutes intervals, significant effect of time and group ($n = 17$ and 15 animals, time: $p = 0.0000092$, treatment: $p = 0.0066$, time x treatment: $p = 0.072$, two-way repeated measure ANOVA). aCB1KO mice activated the air puff more often during the first 5 min compared to sham mice (2.35 ± 0.33 vs. 4.07 ± 0.35, $p = 0.0065$, post-hoc Tukey test) but not during the last 5 min (1.29 ± 0.33 vs. 1.87 ± 0.35, $p = 0.64$, post-hoc Tukey test). Sham mice did not show a difference over time (2.35 ± 0.33 vs. 1.29 ± 0.33, $p = 0.13$, post-hoc Tukey test) while aCB1KO mice did (4.07 ± 0.35 vs. 1.87 ± 0.35, $p = 0.00063$, post-hoc Tukey test). **d** Out of all mice activating the air puff at position B ≤ 3 times ($n = 13$ mice), i.e. mice that avoided the new puff position quickly, 76.9 % were sham ($n = 10$) and 23.1 % aCB1KO mice ($n = 3$) ($p = 0.036$, two-sided Fisher's exact test). Data are expressed and displayed as mean ± s.e.m. or displayed in box plots. The box indicates the 25th and 75th, the whiskers the 5th and 95th percentiles, the horizontal line in the box the median and the mean is represented by a filled circle. Source data are provided as a Source Data file.

anterior cortical astrocytes improves remote object recognition memory[46].

Further studying the role of astrocytic CB1Rs in spatial memory in a passive place avoidance test, we revealed that the spatial memory deficit uncovered in the object location test can be overcome if the relevance of a location is increased by an aversive stimulus, i.e., an air puff. A similar improvement of memory by aversive cues has been demonstrated previously. For instance, the performance of mice in the Morris water maze was increased in the presence of a predator odor, which was shown to involve the amygdala[47]. It is therefore likely that the air puff used in our spatial passive avoidance test enforces spatial learning by involving the amygdala. Interestingly, the spatial memory impairment of mice with CB1R-deficient astrocytes resurfaced transiently when the location of the aversive stimulus was moved (see Fig. 9c). In such experiments that test the association of an aversive stimulus and a context, the hippocampus is believed to encode the context[38]. The transient deficit in acquiring the new location of the air puff is therefore likely of hippocampal origin and therefore associated with the impaired dendritic spiking of CA1 pyramidal cells in these mice in vitro. Whether this is in fact a causal relationship is a challenging experimental question. Another prediction arises from the fact that astrocytic CB1Rs mediate astrocytic Ca²⁺ transients and that its disruption impairs spatial memory. This indicates that some aspect of

the spatial memory task such as the location of the animal should be encoded by astrocytic Ca²⁺ during exploration. Indeed, there is prominent astrocytic Ca²⁺ signaling during locomotion[48–50] and astrocytic activity has recently been used to predict a reward location in a familiar environment[51].

At the level of NMDARs, our current results emphasize that the endogenous supply of the NMDAR co-agonist D-serine is dynamically regulated and that astrocytes and their Ca²⁺ signaling play an important role in setting the availability of D-serine[13–16,23,24]. We demonstrate that astrocytic Ca²⁺ signals initiated by the activation of astrocytic CB1Rs[16,23,25] either pharmacologically or by endogenous pyramidal cell activity led to increased co-agonist levels and thereby a reduced dendritic spike threshold and an increased dendritic spike amplitude. This chain of events could be disrupted in experiments with a D-serine degrading enzyme, indicating that indeed D-serine had been released, using CB1R-deficient astrocytes, which revealed that astrocytic CB1Rs are involved, and astrocytic expression of a Ca²⁺ pump[39] implicating astrocytic Ca²⁺ signaling. Both astrocytes and neurons have been demonstrated to control extracellular D-serine levels but the specific and relative contributions of either cell type remain a matter of debate[52,53]. Although we did not dissect the release mechanisms of D-serine, our results clearly demonstrate that astrocytes are key regulators of extracellular D-serine levels[13–15,18,23,24,27,54,55]. They also add to

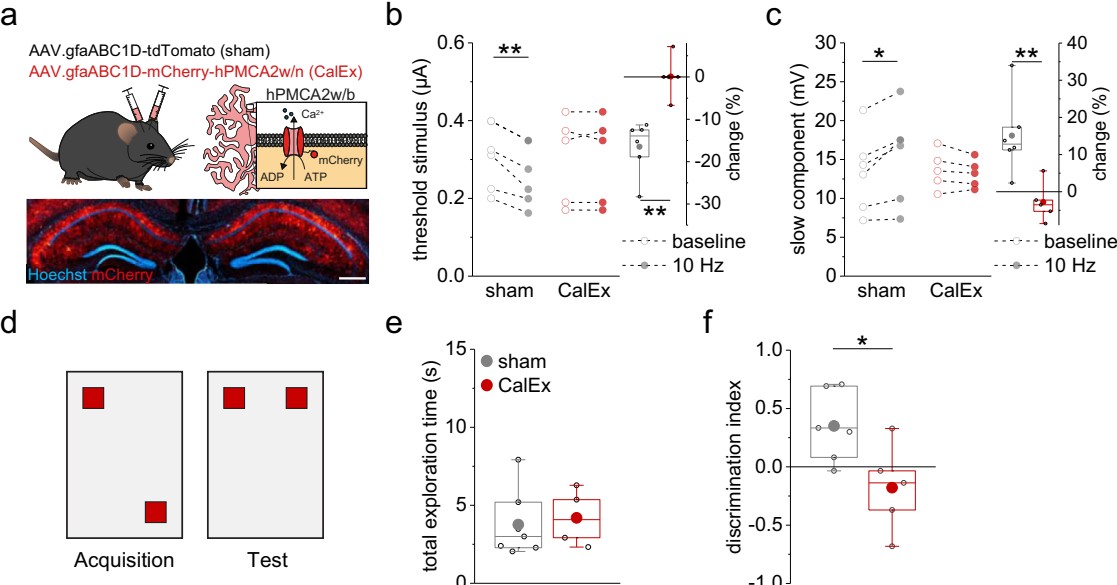

**Fig. 10 | CalEx disrupts the astrocytic feedback loop and formation of spatial memory. a** C57Bl6/N mice were injected bilaterally (CA1, hippocampus), with either AAV.gfaABC1D-tdTomato (sham) or AAV.gfaABC1D-mCherry-hPMCA2w/n (CalEx)[39]. Absence of overlap between mCherry expression and neuronal cell layers (lower panel, scale bar 500 μm, see Supplementary Fig. 9). **b** In sham-injected mice, 10 Hz stimulation decreased the dendritic spike threshold stimulus significantly (left panel; gray: 0.31 ± 0.03 μA vs. 0.26 ± 0.03 μA, $n = 6$, $p = 0.0021$, two-sided paired Student's $t$ test) but not in CalEx mice (red: 0.30 ± 0.05 μA vs. 0.30 ± 0.05 μA, $n = 5$, $p = 1.00$, two-sided paired Student's $t$ test). The change was significantly larger in sham mice (right panel; sham vs. CalEx, −16.47 ± 2.61 vs. 0.10 ± 2.19, $n = 6$ and 5, $p = 0.0011$, two-sided Student's $t$ test). **c** Corresponding results for the slow component (left panel; sham, gray: 13.36 ± 2.06 mV vs. 15.47 ± 2.42 mV, $n = 6$, $p = 0.016$, two-sided paired Student's $t$ test; CalEx, red: 13.66 ± 1.11 mV vs.

13.18 ± 0.79 mV, $n = 5$, $p = 0.23$, two-sided paired Student's $t$ test). Right panel: relative change 10 Hz/baseline (sham vs. CalEx, 15.09 ± 4.29 vs. −2.80 ± 2.36, $n = 6$ and 5, $p = 0.0073$, two-sided Student's $t$ test). **d** Testing arena for object location memory (also Fig. 8). **e** Similar total exploration time during acquisition between groups (sham vs. CalEx: 3.76 ± 0.80 s vs. 4.19 ± 0.74 s, $n = 7$ and 5, $p = 0.71$, two-sided Student's $t$ test). **f** Sham mice explored the object in the novel location significantly more compared to, CalEx mice (0.35 ± 0.11 vs. −0.18 ± 0.17, $n = 7$ and 5, $p = 0.019$, two-sided Student's $t$ test). Data are expressed as mean ± s.e.m. in the legend and displayed in box plots. The box indicates the 25th and 75th, the whiskers the 5th and 95th percentiles, the horizontal line in the box the median and the mean is represented by a filled circle. Source data are provided as a Source Data file.

the growing experimental evidence indicating that astrocytic CB1Rs are an important regulator of a variety of complex behaviors[16,56,57]. In addition, we uncovered a previously unknown mechanism that dynamically adjusts NMDAR co-agonists levels according to neuronal activity.

These observations do not rule out that the NMDAR co-agonist glycine may also play a role for dendritic integration. A previous study has revealed that the co-agonist for synaptic NMDARs is primarily D-serine whereas it is glycine for extrasynaptic NMDARs[28]. The latter are however likely to be activated during glutamatergic synaptic crosstalk in the hippocampus[58]. They also contribute to NMDA spikes in cortical layer V pyramidal cell dendrites[59] and it is conceivable that activation of hippocampal extrasynaptic NMDARs becomes more prominent during the simultaneous activity of spatially clustered synapses that is required for triggering a local dendritic spike. Interestingly, extracellular glycine levels are actively maintained by glycine transporters in the hippocampus[17,60] and we could previously demonstrate that they are modulated by neuronal activity using a newly-designed optical glycine sensor[61]. An interesting open question is therefore if activity-dependent changes of extracellular glycine levels also control dendritic spiking. Similarly, the spine-to-spine variability of glutamate uptake efficiency[62] and the increased glutamatergic crosstalk due to withdrawal of perisynaptic astrocyte processes from synapses undergoing long-term potentiation[63] may profoundly affect dendritic spike generation, their dependence on NMDAR co-agonists and place field formation. Beyond NMDAR co-agonists, it will also be interesting to see if the frequency-dependent increase of astrocytic Ca²⁺ signals also triggers a frequency-dependent release of other signaling molecules from astrocytes such as GABA and/or ATP[18,64,65]. Similarly, the quantitative and spatial relationship

between astrocytic Ca²⁺ increases and the release and spread of released signaling molecules remains to be fully established.

## Methods

All research and experimental procedures complied with the relevant ethical regulations. All procedures involving animals were conducted in accordance with the regulations of the European Commission and all relevant national and institutional guidelines and requirements (e.g. Haus für Experimentelle Therapie [HET], University Hospital Bonn, Germany). Procedures have been approved by the Landesamt für Natur, Umwelt und Verbraucherschutz Nordrhein-Westfalen (LANUV, Germany) where required.

### Animals

All animals were housed under 12 h light/dark conditions and were allowed ad libitum access to food and water (room temperature 22 °C, air humidity 50-70 %). The experiments were performed using 3−5-week-old male Wistar rats (Charles River), 6-10-week-old male C57BL/6 N wild type (Charles River) and transgenic mice of the following lines: GLAST-creERT2[33], CB1fl/fl (Marsicano et al., 2003), flox-top tdTomato[35] (JAX 007908), flox-stop GCaMP5g-IRES-tdTomato[66] (JAX 024477). All transgenic mouse lines were bred to have an C57BL/6 N background. Transgenic mice of both sexes were used to minimize breeding. To induce cre expression, 3-week-old mice were injected with tamoxifen (100 mg / kg BW [body weight]; 1/day i.p., 5 days).

### Stereotactic injections

For stereotactic injections of rAAVs into the CA1 region of the hippocampus, flox-stop GCaMP5g-IRES-tdTomato or C57BL/6 N mice were deeply anesthetized (Fentanyl, Rotexmedica, 0.05 mg/kg bodyweight;

Midazolam, Rotexmedica, 5.0 mg/kg bodyweight; Medetomidin, Cepitor from CPPharma, 0.5 mg/kg bodyweight; injection volume 0.1 ml/10 g bodyweight, i.p.). The eyes were covered with a crème (Bepanthen® eye and nose crème), and the anesthesia was confirmed by testing toe reflexes. Next, the head fur was shaved, the head was fixed in a stereotactic frame (Model 901, David Kopf Instruments) and the skin was locally anaesthetized (1 × 10 mg puff, Xylocain, Astra Zeneca). Then, a small incision was made, and the skull was carefully cleared from the remaining periosteum. A small whole (coordinates relative to *bregma*: ventral CA1: anterior −3.5 mm, lateral ±3 mm, ventral −2.5 mm; dorsal CA1: anterior −1.8 mm, lateral ±1.3 mm, ventral −1.6 mm) was drilled with a dental drill, and a beveled needle nanosyringe (nanofil 34 G BVLD, WPI) was slowly inserted into the brain. After allowing the tissue to adjust, 0.5 to 1 µl of viral particles (VPs; $2.94 \times 10^{13}$ VP/ml, AAV1.CaMKII0.4Cre.SV40, V4567MI-R, PennCore; $3.2 \times 10^{12}$ VP/ml, AAV5.gfaABC1D-cyto-GCaMP6f, 52925, Addgene; $7 \times 10^{12}$ VP/ml, AAV5.gfaABC1D-mCherry-hPMCA2w/b, 111568, Addgene; $1 \times 10^{12}$ VP/ml, AAV1/2-gfaABC1D-tdTomato, see[67]) were injected under the control of a microinjection pump (100–200 nl/min, Micro4 Microsyringe Pump Controller, WPI). The needle was left in place for at least five minutes to avoid reflux into the needle track. Then, the needle was retracted slowly, and the procedure was repeated for the other hemisphere. Finally, the incision was sutured (anti-bacterial absorbable thread, Ethicon) and an antibiotic was applied (Refobacin 1 mg/g, Gentamicin). The anesthesia was terminated (Naloxon, Puren, 1.2 mg/kg bodyweight; Flumazenil, Anexate from Hikma, 0.5 mg/kg bodyweight, Atipamezol, Antisedan from Ventoquinol, 2.5 mg/kg bodyweight; injection volume 0.1 ml/10 g bodyweight, i.p.) and the mouse was placed back into its home cage. Analgesia was applied 30 min before terminating the anesthesia as well as in case of buprenorphin 8, 16 and 24 h after the surgery (Buprenovet from Bayer, 0.05 mg/kg bodyweight; injection volume 0.1 ml/20 g bodyweight, i.p.) and in case of carprofen 24 h after surgery (Rimadyl from Zoetis, 5 mg/kg bodyweight; injection volume 0.1 ml/20 g bodyweight, s.c.). These animals were used for experiments three to five weeks after surgery.

## Hippocampal slice preparation

Acute brain slices were prepared as previously described[68] in the late morning, i.e., in the beginning of the non-active light phase of the animals. Briefly, 300 µm slices were prepared in an ice-cold slicing solution (in mM: sucrose 105, NaCl 60, KCl 2.5, $MgCl_2$ 7, $NaH_2PO_4$ 1.25, ascorbic acid 1.3, sodium pyruvate 3, $NaHCO_3$ 26, $CaCl_2$ 0.5 and glucose 10; osmolarity 300–310 mOsm/l). After a recovery of 15 min in 34 °C-warm slicing solution, the slices were stored in artificial cerebrospinal fluid (ACSF, composition in mM: NaCl 131, KCl 2.5, $MgSO_4$ 1.3, $NaH_2PO_4$ 1.25, $NaHCO_3$ 21, $CaCl_2$ 2 and glucose 10; pH 7.35–7.45; osmolarity 297–303 mOsm/l) for at least one hour before the start of the experiment. All experiments were conducted at ~34 °C.

## Glutamate iontophoresis

Acute slices were transferred into a submersion-type recording chamber mounted on a Scientifica two-photon excitation fluorescence microscope with a 40x/0.8 NA objective or 60x/1 NA objective (Olympus), placed onto a self-build grid and superfused with ACSF at 34 °C. CA1 pyramidal cells were patched in the whole-cell configuration using a Multiclamp 700B amplifier (Molecular Devices, pClamp 10.3) with borosilicate glass pipettes (3–5 MΩ resistance, GB150F-10, Science Products) filled with an intracellular solution containing in mM: $KCH_3O_3S$ 135, HEPES 10, di-Tris-Phosphocreatine 10, $MgCl_2$ 4, $Na_2$-ATP 4, Na-GTP 0.4, Alexa Fluor 594 hydrazide 0.04 (Thermo Fisher Scientific), Fluo-4 pentapotassium salt 0.2 (Thermo Fisher Scientific), pH adjusted to 7.2 using KOH, osmolarity 290–295 mOsm/l. Cells were kept in the current clamp mode. Using a Ti:sapphire pulsed laser (Vision S, Coherent) the fluorescent indicators were excited

($\lambda = 800$ nm). A glutamate microiontophoresis system (MVCS-C-01C-150, NPI) was used for local dendritic activation. For this purpose, a borosilicate glass pipette (60-90 MΩ resistance, GB150F-10, Science Products), filled with 150 mM glutamic acid (pH adjusted to 7.0 with NaOH) and 50 µM Alexa Fluor 594 hydrazide (Thermo Fisher Scientific), was placed in close proximity (<1 µm) to a spine on an apical oblique dendrite under the guidance of two-photon excitation fluorescence microscopy. We chose mostly dendrites directly branching off the apical trunk dendrite, which were stimulated at about 2/3 of their length from the apical trunk dendrite. Leakage of glutamate was prevented by a small, positive retain current (<8 nA). The iontophoretic stimulation duration was adjusted (<0.8 ms) so that reliable somatic excitatory postsynaptic potentials (EPSPs), dendritic spikes and action potentials (APs) could be recorded with increasing stimulation intensity (25 to 50 pA steps until the occurrence of an AP, 3 s interstimulus interval). All experiments were conducted in the presence of 50 µM picrotoxin (abcam Biochemicals) to inhibit $GABA_A$ receptors. Only cells with an initial access resistance ($R_a$) <20 MΩ and with <20 % change during the time course of the recording were included in the analysis.

## Glutamate uncaging

Acute slices were transferred to a submerged recording chamber on a dual galvanometer based scanning system (Prairie Technologies). CA1 pyramidal cells were visualized with infrared oblique illumination optics and a water immersion objective (60x, 0.9 NA, Olympus) and somatic whole-cell current clamp recordings were performed with a BVC-700 amplifier (Dagan Corporation). Data were filtered at 10 kHz and sampled at 50 kHz with a Digidata 1440 interface controlled by pClamp 10.3 (Molecular Devices). Patch-pipettes were pulled from borosilicate glass (outer diameter 1.5 mm, inner diameter 0.8 mm; Science Products) with a P-97 Puller (Sutter Instruments) to resistances of 2 to 5 MΩ in bath and whole-cell series resistances ranging from 8 to 30 MΩ. The standard internal solution contained (in mM): 140 K-gluconate, 7 KCl, 5 HEPES, 0.5 MgCl2, 5 phosphocreatine, 0.16 EGTA. Internal solutions were titrated to pH 7.3 with KOH, had an osmolality of 295 mOsm/l, and contained 100 µM Alexa Fluor 594 (Thermo Fisher Scientific). Voltages were not corrected for the calculated liquid-junction potential of +14.5 mV. The membrane potential was adjusted to −75 mV for all recordings. Cells with unstable input resistances or lacking overshooting action potentials were discarded as well as recordings with holding currents > −200 pA at −60 mV and access resistances >30 MΩ. For two-photon glutamate uncaging at apical oblique dendrites of CA1, MNI-caged-L-glutamate 15 mM (Tocris) was dissolved in HEPES-buffered solution (in mM as follows: 140 NaCl, 3 KCl, 1.3 $MgCl_2$, 2.6 $CaCl_2$, 20 D-glucose, and 10 HEPES, pH 7.4 adjusted with NaOH, 305 mOsm/l) and applied using positive pressure via glass pipettes (<1 MΩ) placed in close proximity to the selected apical oblique dendrites of CA1 neurons. We used two ultrafast laser beams of Ti:sapphire pulsed lasers (Chameleon Ultra, Coherent), which were tuned to 860 nm to excite the Alexa 594 and to 720 nm to photorelease glutamate at 10-15 dendritic spines of a dendritic segment of ~10 µm in length. The intensity of each laser beam was independently controlled with electro-optical modulators (Conoptics Model 302RM). MNI-glutamate was uncaged at an increasing number of spines (2-15) with 0.5 ms exposure times and the uncaging spot was rapidly moved from spine to spine with a transit time of ~0.1 ms. The laser power at the slice surface was kept below 22 mW to avoid photo damage. Glutamate was uncaged onto a sequence of single spines to evoke excitatory post synaptic potentials (uncaging evoked EPSPs, uEPSPs). To quantify deviations from linearity in dendritic integration, the arithmetic sum of individual uEPSPs of single spines (expected EPSP) was compared to the measured EPSP evoked by rapid sequential glutamate uncaging onto the same set of spines. The rate of rise of the dendritic spike's initial fast phase was calculated from the maximum dV/dt value.

The amplitude of the slow phase of dendritic spikes was obtained from the same dendritic spikes used to quantify dV/dt. The dendritic spike threshold was calculated as the amplitude of the expected EPSP at which dendritic spikes first occurred. All data analyses were done with pClamp 10.3 (Molecular Devices) and IGOR Pro 9.0 (Wavemetrics).

## Alveus stimulation

Selective activation of axons in the alveus was achieved through an incision between CA1 and the subiculum that spared the alveus. Next to the subicular cut, a clustered bipolar stimulation electrode (CE2F75, FHC) was placed (Supplementary Fig. 2a). A borosilicate glass pipette (2–4 MΩ resistance, GB150F-10, Science Products) filled with ACSF was positioned at the boarder of stratum oriens and stratum pyramidale of CA1 to record the evoked population spikes. The stimulation intensity was adjusted to 80% of that evoking the maximum population spike amplitude. Only slices with a minimum population spike amplitude of 0.8 mV were used for experiments. In a subset of experiments the glass pipette was removed and a CA1 pyramidal cell within the same area (±70 μm) was patched and stimulated using glutamate iontophoresis as described above. In another set of recordings, extracellular single unit responses were recorded. In these recordings, a borosilicate glass pipette (5–7 MΩ resistance, GB150F-10, Science Products) was placed into the pyramidal cell layer during 10 Hz alveus stimulation to blindly identify single unit responses. These responses are characterized by a sharp upward reflection followed by a smaller downward reflection. Next, recordings at different frequencies (4, 10 and 40 Hz) of alveus stimulation were done in quadruplets. A maximum of three single unit recordings per slice were performed.

## Field recordings in acute hippocampal slices

In a subset of experiments (Supplementary Fig. 7a, b), field EPSPs (fEPSPs) were evoked by electrical stimulation of CA3-CA1 Schaffer collaterals (100 μs) and recorded in the CA1 stratum radiatum through an extracellular pipette (patch pipette as described above) filled with the extracellular solution. For the characterization of synaptic transmission, the stimulus intensity was increased in a stepwise manner from 20 to 640 μA. For paired-pulse experiments, the stimulus intensity was set to obtain a half-maximal response and the inter-stimulus interval was varied. In another subset of experiments (Supplementary Fig. 7c), an extracellular recording electrode was placed in the pyramidal cell layer to record the population spike after synaptic stimulation. Again, the stimulus intensity was increased in a stepwise manner from 20 to 640 μA to compare population spikes and their dependence on the strength of stimulation between experimental groups.

## Pharmacology

The following drugs in the respective concentrations were used: 5 μM AM251 (abcam), 0.17 U/ml DAAO (Sigma), 50 μM D-APV (abcam), 10 μM DCKA, 10 μM D-serine (Sigma), 10 μM NBQX (abcam), 1 μM WIN55,212-2 mesylate (Sigma), 10 μM ZD7288 (Tocris).

## Astrocyte Ca²⁺ imaging

Detection and analysis of astrocytic $Ca^{2+}$ signaling was performed using the approaches described in the two sections below. In several experiments, we analyzed large regions of interest (ROIs) covering the entire astrocyte but sparing the soma and major branches. This was done to focus on $Ca^{2+}$ signals in the periphery of the astrocytes, presumably closer to synapses, and to avoid defining complex criteria for event detection or ROI placement and thus to also avoid the potential biases associated with these[69]. As a consequence, $Ca^{2+}$ response amplitudes are averaged over large regions of interest in these analyses and can therefore be lower than local maximum values in the analyzed region. Throughout we isolated "green" and "red" indicator fluorescence by a dichroic mirror and band pass filters (Scientifica

microscope: 565 nm dichroic, 500–550 nm, 590–650 nm; Olympus microscope: 570 nm dichroic, 515–560 nm, 575–630 nm).

## Astrocyte whole-cell patch clamp and Ca²⁺ imaging

Detailed explanations of experimental procedures were described previously[70,71]. Acute slices were transferred to a submersion-type recording chamber mounted on a Scientifica two-photon excitation fluorescence microscope with a 40x/0.8 NA objective (Olympus). A stratum radiatum astrocyte at a depth of 40–60 μm was recorded from in the whole-cell patch clamp configuration using a Multiclamp 700B amplifier (Molecular Devices) using borosilicate glass pipettes (3.0–3.5 MΩ resistance, GB150F-10, Science Products) using an intracellular solution containing in mM: $KCH_3O_3S$ 135, HEPES 10, di-Tris-Phosphocreatine 10, $MgCl_2$ 4, $Na_2$-ATP 4, Na-GTP 0.4, Alexa Fluor 594 hydrazide 0.04 (Thermo Fisher Scientific), Fluo-4 pentapotassium salt 0.2–0.4 (Thermo Fisher Scientific), pH adjusted to 7.2 using KOH, osmolarity 290–295 mOsm/l and kept in the current-clamp mode. Fluorescent dyes were excited with a Ti:sapphire pulsed laser (Vision S, Coherent, λ = 800 nm). The cellular identity was confirmed by their membrane potential (<−80 mV), their low input resistance (<5 MΩ), their typical morphology and gap junction coupling (see Supplementary Fig. 11 for an example). Recordings in which the access resistance exceeded 30 MΩ were excluded from the analysis.

In experiments from Fig. 3a, b, time-lapse frame scanning (0.75-1.5 Hz) was performed after sufficient dye diffusion (>20 min) during baseline and after bath application of 10 μM of WIN 55,212-2. A subset of these experiments was performed in the presence of the inverse endocannabinoid receptor agonist AM251 (5 μM). For analysis, the fluorescence intensity ratio (R, Fluo-4 / Alexa 594, both channels corrected for background fluorescence) was quantified in a single region of interest (ROI) covering the entire astrocyte sparing the soma and major branches during WIN55 application (R) and before ($R_0$). The effect of WIN55 was quantified by calculating $R/R_0$.

In experiments from Fig. 4a–c, alveus stimulation was combined with astrocyte whole-cell patch clamp recordings and $Ca^{2+}$ imaging. An alveus stimulation electrode was positioned, the stimulation intensity was set as described and an astrocyte was patched. After sufficient dye diffusion (>20 min), time lapse imaging was performed during baseline, during alveus stimulation (3 × 20 Hz for 1 s), and after bath application of the inverse endocannabinoid receptor agonist AM251 (5 μM) during a second round of alveus stimulation. The overall change of astrocytic $Ca^{2+}$ signals was estimated by first calculating the ratio of both fluorescent indicators (R, Fluo-4/Alexa Fluor 594, botch channels corrected for background fluorescence) in ROIs covering the entire astrocytic territory sparing the soma and major branches. For each step of the recording (baseline, alveus stimulation, alveus stimulation in AM251), the resting value $R_0$ and the average $\Delta R$ (change relative to $R_0$) were determined and $\Delta R/R_0$ was calculated as a measure of overall $Ca^{2+}$ signal activity. $\Delta R/R_0$ during stimulation/treatment was compared to the baseline period to quantify the effect of stimulation/treatment (Fig. 4c, right panel). ROIs that showed an increase of $\Delta R/R_0$ of more than 5% during alveus stimulation were defined to be responders (Fig. 4c, left panel) and used to analyze the effect of AM251.

## Calcium imaging using genetically encoded Ca²⁺ indicators

The expression of GcaMP5g and tdTomato in astrocytes or CA1 pyramidal cells was established through crossbreeding flox-stop GcaMP5g-IRES-tdTomato mice with GLAST-creERT2 mice and intra-peritoneal tamoxifen injections (astrocytes) as described before[50] or rAAV injections (pyramidal cells, see above). Horizontal hippocampal slices were prepared and recorded from on an Olympus FV10MP microscope with a 25x/1.05NA objective or a Scientifica two-photon excitation fluorescence microscope with a 40x/0.8 NA objective at 34 °C in ACSF (supplemented with 50 μM Picrotoxin as in all other acute slice experiments). Using a Ti:sapphire pulse laser

(Vision S, Coherent) both fluorescent proteins were excited ($\lambda = 910$ nm). Acquisition of the images was performed using Scan-Image 3.8 or Olympus FluoView 4.2.

For experiments in Supplementary Fig. 2 on CA1 pyramidal cells, the response of up to six CA1 pyramidal cells to 20 Hz alveus stimulation was determined by line scanning (~2 ms per line, 1500 lines). The intensity profile of the fluorescent ratio (GcaMP5g/tdTomato, background corrected) was analyzed. Recordings were performed in quadruplets for at least 16 cells per slice to determine the percentage of responding cells. An arbitrary threshold of 25% well above the noise level was defined to identify responding cells.

In experiments from Fig. 5a–c, astrocytic calcium transients were recorded using time-lapse frame scanning (128 ×128 pixels, 80 µm x 80 µm, 2.96 Hz) during baseline and 10 Hz or 40 Hz alveus stimulation for three minutes each. The overall changes of astrocytic $Ca^{2+}$ signals were estimated by first calculating the ratio of both fluorescent indicators (R, GcaMP5g/tdTomato, both channels corrected for background fluorescence) in single ROIs covering the entire astrocyte territory sparing the soma and major branches. For each period of the recording (baseline, alveus stimulation at 10 or 40 Hz), the resting value $R_0$ and the average $\Delta R$ (change relative to $R_0$) were determined and $\Delta R/R_0$ was calculated as a measure of overall $Ca^{2+}$ signal activity. $\Delta R/R_0$ during stimulation was compared to the baseline period to quantify the effect of stimulation (Fig. 5c). ROIs that showed an increase of $\Delta R/R_0$ of more than 5% during alveus stimulation were defined to be responders (Fig. 5b).

In analyses in Supplementary Fig. 4, the data set from Fig. 5a–c was re-analyzed by manual identification of $Ca^{2+}$ transients. For each cell, $Ca^{2+}$ transients were visually identified, counted, and the frequency of events during the baseline period and alveus stimulation was calculated (Supplementary Fig. 4b–d). For the analysis of $Ca^{2+}$ signal amplitudes, the fluorescence ratio of both indicators (R, GcaMP5g/tdTomato, both channels corrected for background fluorescence) was determined immediately before each $Ca^{2+}$ transient ($R_0$) and at its peak to obtain its change ($\Delta R$) in ROIs of 3 µm x 3 µm. The $Ca^{2+}$ transient amplitude was calculated as $\Delta R/R_0$. The mean amplitude of astrocytic $Ca^{2+}$ transients was then compared between baseline recordings and alveus stimulation per cell (Supplementary Fig. 4e).

In experiments from Fig. 5f, g and Supplementary Fig. 5, the expression of GcaMP6f in astrocytes was obtained through rAAV injections. Horizontal hippocampal slices were prepared and recorded from on an Olympus FV10MP microscope with a 25x/1.05NA objective at 34 °C in ACSF (supplemented with 50 µM picrotoxin as in all other acute slice experiments). Using a Ti:sapphire pulsed laser (Vision S, Coherent) GcaMP6f was excited ($\lambda = 910$ nm). Acquisition of the images was performed using Olympus Fluoview 4.2. Astrocytic $Ca^{2+}$ events of one to four cells were recorded under control conditions or in the presence of 10 µM ZD7288 using time-lapse frame scanning (pixel size 0.24 – 0.44 µm/px, frame rate 0.61–1.79 Hz) during baseline and 10 Hz alveus stimulation for ten minutes each. After motion correction of the recordings using NoRMCorre[72], the event-based detection algorithm AquA[73] was used for detecting astrocytic $Ca^{2+}$ events. Signal detection and z-score thresholds were adjusted for each recording to account of differences in noise level. Somatic events, events with a duration of less than 3 frames and events with a negative decay time constant were excluded.

### Immunohistochemistry

Mice were anesthetized using an overdose of ketamine and xylazine, the chest cavity opened and transcardially perfused with at least 20 ml ice-cold paraformaldehyde (4% in PBS). After removal of the brain, it was placed in the same solution overnight for post-fixation. Next, the brain was cut in 50 µm thick, horizontal slices using a vibratome (Leica). After blocking for 2 h in 10 % normal goat serum (NGS, Merck Millipore) and 0.5% Triton (AppliChem) in PBS, the slices were stained with primary antibodies for GFAP (polyclonal rabbit anti-cow GFAP, 1:500; Z0334, Dako; polyclonal chicken anti-human GFAP, 1:500, 173 006, Synaptic Systems) and NeuN (monoclonal mouse anti NeuN, clone A60, 1:200; MAB377, Chemicon) in PBS with 5 % NGS and 0.1 % Triton overnight at 4 °C. After washing, the slices were places in in PBS with 2 % NGS, 0.1 % Triton and the secondary antibodies for anti-rabbit (polyclonal goat anti-rabbit IgG (H + L)-Alexa Fluor 488, 1:500; A11034, Invitrogen), anti-chicken (polyclonal goat anti-chicken IgG (H + L)-Alexa Fluor 488, 1:500; A11039, Invitrogen) and anti-mouse (polyclonal goat anti-mouse IgG (H + L)-biotin, 1:500; 115-065-003, Dianoca) for one to one and a half hours at room temperature. This was followed by a streptavidin conjugate to fluorescently tag biotin (Alexa Fluor 647 streptavidin conjugate, 1:600; S32357, Thermo Fisher Scientific). For nuclear staining, the slices were placed in water with 0.5 % Hoechst (33342, ThermoFisher Scientific) for ten minutes at room temperature. Finally, the slices were mounted onto microscope slides (Thermo Scientifica) with a ProLong Gold antifade mountant (Invitrogen) and stored at 4 °C. For quantification of specificity and sensitivity of the aCB1KO mice, image stacks of the hippocampal CA1 area (512 × 512 µm, 450 × 450 pixel, pixel size 1.14 µm, 2 µm z-steps) were taken using a laser scanning confocal microscope (TCS SP8, Leica). Three sections of 20 µm depth each were averaged per animal for analysis. Cells were counted by the nuclear staining.

### Behavioral tests

GLAST-creERT2 mice[33] were crossed with CB1[fl/fl] mice[34] and flox-stop tdTomato mice[35]. After weening, mice of both genders were injected with either tamoxifen or a sham solution and housed under a reversed 12 h light/dark conditions. In another set of experiments, C57BL6/N mice (Charles Rivers) underwent stereotactic injections, as described, with either AAV.gfaABC1D-tdTomato[67] or AAV.gfaABC1D-mCherry-hPMCA2w/n[39]. After one week of post-operative observation, the mice were housed under reversed 12 h light/dark conditions.

Handling and behavioral experiments as well as tamoxifen/stereotactic injection and analysis were performed by individual, blinded experimenters. Behavioral testing was performed in a separate, quiet area during the beginning of the dark phase. The room was lit with dimmed red light and spatial cues were present around the arena. All test and training sessions were recorded using a Basler acA1300-200um camera system mounted above the arena (1280 × 1024 pixels). After each individual animal test, the arena was carefully cleaned using 70% ethanol.

Habituation (3–5 days) and the open field test (first day of habituation) took place in a rectangle arena (40 × 60 cm) that was constructed of dark gray PVC with 23 or 44 cm wall height and light-gray flooring. The mice were tracked automatically by three points (head, body, tail), manually checked and if needed adapted using EthoVision XT 14.0 (Noldus). For analysis of anxiety-related behavior an outer zone of an eight cm wide gallery and an inner zone of 24 ×44 cm were defined.

The same arena was used for testing object location memory. Two identical objects (4 cm in diameter) were placed randomly in the corners of the arena with a distance of 9 cm from the walls. The mice were placed in the middle of the arena and allowed to explore the objects freely for 10 min. After a 24 h delay, one of the objects was placed to a different corner and the mice were placed back into the arena for five minutes of exploration. Tracking and analysis were performed with EthoVision XT 14.0 (Noldus). Object exploration was defined as the nose-point being within 2 cm of the object, with body orientation towards the object. The discrimination index ((time at object with novel location−time at object with constant location) / total time) was used as an indicator of the object location memory. Animals that showed excessive climbing or little exploration (exploration time <5 s in 10 min or <2 s in 5 min; total distance <1 m in 5 min) as well as outliers for the analyzed parameters (mean ±2 SD) were excluded (tamoxifen

injected mice: $n = 5$ out of 37 mice; virus injected mice: $n = 9$ out of 21 mice).

Spatial working memory was investigated using a Y-maze with three identical arms, each 40 cm long and 8 cm wide, interconnected at 120° angle. Each arm had a different spatial cue at the end. The mice were placed in the middle of the arena and were allowed to explore it freely for ten minutes. Tracking (one point, body) and analysis was performed with EthoVision XT 14.0 (Noldus). The number and order of entrances into the second half of an arm was determined automatically. The number of performed alternations (triplet of visits to all three arms) was divided by the number of possible alternations to calculate the alternation index. Animals were considered to have an intrinsic arm preference or aversion when they entered an arm 10 % more or less often than expected (33%, relative arm visits <23% or >43%). Those animals were excluded from further analysis ($n = 3$).

The passive place avoidance test was performed in a modified open field arena with a 10 cm wide gallery. Light sensors on each short side of the gallery (position A and B) were used to activate an air puff (one bar, nitrogen gas) when the animal was disrupting the light bridge. The animal was placed in the long arm of the arena in equal distance to the possible air puff locations. Each day a single trial of free exploration (10 min) was performed. The animal's behavior was tracked using EthoVision XT 14.0 (Noldus). Activation of the light bridge was determined through entry of the nose-point into a small area (1 cm wide) at the air puff location.

### Image analysis
Image analyses were performed as described above using FIJI (2.3.0) and Matlab 2020 (Mathworks).

### Statistics
Statistical analyses were performed with OriginPro 2017 (OriginLab) and Matlab 2020 (Mathworks). Data are expressed and displayed as mean ± s.e.m. (standard error of mean) or in box plots. In the latter, the box indicates the 25th and 75th, the whiskers the 5th and 95th percentiles, the horizontal line in the box the median and the mean is represented by a filled circle. Statistical tests were performed after investigating whether data followed a normal distribution (Shapiro-Wilk test). Depending on the outcome, statistical tests were performed using non-parametric (Kruskal-Wallis test, Friedman test, Mann–Whitney U-test, Wilcoxon signed rang test) or parametric (ANOVA, repeated measures ANOVA, Student's $t$-tests) approaches as indicated. Probability distributions were compared with the Kolmogorov-Smirnov test. All tests are two-sided. In graphs, statistical significance is indicated by asterisks. * for $p < 0.05$, ** for $p < 0.01$ and *** for $p < 0.001$. Please see Supplementary Table 2 for a comprehensive overview of statistical tests and their results. Examples and representative images were taken from the data set illustrated in the associated panels and described in the figure legend.

### Reporting summary
Further information on research design is available in the Nature Portfolio Reporting Summary linked to this article.

## Data availability
The complete data set generated in this study is described and provided in this document, in the Supplementary Information and the Source Data file. Source data are provided with this paper. Further raw data supporting the current study have not been deposited in a public repository because of their highly diverse nature and formats. They are available from the lead contact upon request. Source data are provided with this paper.

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

## Acknowledgements

We thank Dr. Kurtulus Golcuk, Elisa Fernández and Rebekka Zölzer for assistance with behavioral tests. Research was supported by the NRW-Rückkehrerprogramm (C.H.) and the German Research Foundation (DFG; SFB1089 B03, SPP1757 HE6949/1, FOR2795 HE6949/4, and HE6949/3 to C.H., SFB 1089 C04 and SPP 2041 to H.B.). We were also supported by the Bonn Technology Campus and its Viral Core Facility (VCF) of the Medical Faculty of the University of Bonn.

## Author contributions

K.B. and C.H. designed, performed, and analyzed experiments involving iontophoretic glutamate application. N.M. and H.B. planned and carried out tests using glutamate uncaging. $Ca^{2+}$ imaging was done by K.B. and E.M.S., K.B. and K.H. performed additional electrophysiological tests. Behavioral tests were designed, performed, and analyzed by K.B., A.N.H., A.Z., H.B., and T.O. C.H. conceived the study, planned experiments, analyzed data, and wrote the manuscript, which was subsequently contributed to by all the authors.

## Funding

## Competing interests

The authors declare no competing interests.
