## [Peer Review File · Nature Communications]

REVIEWER COMMENTS

Reviewer #1 (Remarks to the Author):

The paper by Bohmbach et al. explores the role for hippocampal astrocytes and astrocyte-derived D-Serine in the integration of glutamatergic activity in the dendrites of pyramidal neurons. By combining the whole-cell electrophysiological recordings, two-photon microscopy, and astrocyte-specific genetic manipulations, Authors provide several lines of evidence that extracellular level of NMDAR co-agonist determines the threshold of dendritic spiking of CA1 pyramidal cells and that endogenous release of NMDAR co-agonist (most likely D-Serine) critically depends on endocannabinoid Ca²⁺-signalling in astrocytes. Authors also demonstrate that this feed-back mechanism of bi-directional neuron-glia communication depends on the firing frequency of pyramidal neurons.

The paper provides new insights into the mechanisms of astrocyte-neuron communications and would bring a major contribution into current debate on physiological relevance of astrocyte for cognitive functions. Firstly, Authors provide another evidence of importance of astrocytes as a main source of NMDAR co-agonists. Secondly, the presented in situ data on the activation of astrocytic signalling primarily in response to CA1 neurons firing in the theta frequency range could explain many inconsistencies in reports of ability vs. inability of hippocampal astrocytes to respond to physiological levels of synaptic activity.

Finally, the presented behavioural data obtained in the mice with conditional astrocyte-specific deletion of CB1 receptors clearly demonstrate the physiological relevance of astroglial endocannabinoid signalling and astrocyte-neuron interactions for the learning and memory. The paper looks to be suitable for publication in the Nature Communications, the presented results will surely attract a great interest among wide readership of the journal. Yet, there are several issues which need to be clarified to substantiate the message and conclusions of the work.

Major points:

- 1) The suggested mechanism of glia-driven modulation of dendritic integration is indirect and basal levels of occupancy of NMDAR Glycine-site will affect the dynamic range of this regulatory cascade. To verify that this cascade of glia-neuron interaction occurs via modulation of NMDAR Gly-site, it is important to show that antagonists of this site (such as CGP 78608 or L-701,324) block the effect of astrocytic CB1Rs on dendritic spike threshold and amplitude (as in Figure 2).
- 2) The data shown in the Fig.3 and S3, strongly support the role of D-Serine in modulation of dendritic spike firing. These data, however, do not rule out putative release of D-Serine from neurons. Although previous works of the Authors argue very strongly against neurons as a main source of D-Serine, in view of current debate in the field, it is important to substantiate the dominant role for astrocytes. This could be done by demonstrating that intracellular perfusion of astrocytes with Ca-chelators strongly attenuates the effect of alveus stimulation on spike threshold and slow component amplitude.
- 3) Theoretically, there are several alternative mechanisms by which astrocytes could influence neuronal excitability and dendritic integration. Astrocytes can affect the GABAergic inhibition by releasing GABA [Lee, Yoon et al. Science 2010] or via release of ATP causing down-regulation of GABAA receptors [Lalo et al. PloSBiology 2014]. Astrocytes were also shown to synchronize neuronal firing [Pascual 2005] and affect trafficking of postsynaptic AMPA receptors by release of ATP/adenosine [Pouget 2014 Neuron, Rodrigues 2015]. Could Authors rule out the putative involvement of these mechanisms in astroglial control of dendritic integration?
- 4) The intracellular Ca²⁺-elevation in astrocytes is instrumental for the glia-neuron interactions reported in the paper. There is a growing consensus in the field that majority of physiologically-important Ca²⁺-events occur in microdomains in fine astrocytic processes. In this regard, the paper rightfully focuses on Ca²⁺-signalling in astrocytic branches, in particular with the aid of two-

photon fluorescent recordings of GCamp6 signalling. Still, many details of Ca²⁺-measurements need clarification to allow wider audience fully appreciate the importance of CB1R-mediated astroglial signalling:

- (i) Although the Methods mention evaluation of the OVERALL changes in astrocytic Ca²⁺ signals (lines 757, 765), further description and illustrations suggest that Ca²⁺-signalling was evaluated in just one (Figs 2 and 3) or few (Fig.S4) ROIs per cell. Do the values of R/R₀ ratio reported in Figs. 2b and 3c refer to the whole astrocytes or just selected "responder" ROIs? The similar question is applicable to the frequency of Ca²⁺-transients reported in the Fig.S4;
- (ii) To verify that reported data represent the general pattern of changes in astrocytic signalling, Authors need to specify how ROIs were defined and selected for analysis, how many ROIs were analysed per single astrocytes and how the average number of "responders" per cell changes upon stimulation of alveus and application of CB1R modulators;
- (iii) Regarding the signalling microdomains, has any kind of spatial analysis of local elevations in fluorescence (e.g. particle analysis) been performed prior to selection of ROIs ?
- (iv) Overall magnitude of Ca²⁺-elevation: Average amplitude of Ca-transients elicited in astrocytes by alveus stimulation and measured with Fluo-4 (about 25-30% of baseline, Fig.3) appears to be considerable smaller than of transients assessed with genetically-encoded GCamp5d (about 50-70% of baseline, Fig.S4), which might be explained either by buffering of Ca²⁺ with high concentration of Fluo-4 or by preferential expression of GCamp in the vicinity of cell membrane (or both). In Authors opinion, which approach provides the better estimate of cytosolic Ca²⁺-elevation ? Also, do Authors believe that elevation of 30-50% from the baseline Ca²⁺-level is enough to activate a significant release of D-Serine?

5) Spatial specificity of the reported feedback mechanism is a bit puzzling and needs further discussion: The release of endocannabinoids from CA1 neurons onto stratum radiatum astrocytes was caused by stimulation of alveus. Would stimulation of the Schaffer collaterals result in similar Ca²⁺-elevation and release of D-Serine ?

Minor/technical issues:

- 1) technical issues: (i) at what wavelengths the Fluo-4 and Alexa594 signals were detected ? - to ensure the putative cross-bleed between red and green channels did not affect the "responder" detection.
- 2) In the whole-cell experiments in astrocytes, their identity was conferred by the presence of the gap-junction coupling. How this was assessed, via spreading of fluorescent dye or by paired electrophysiological recordings? Showing some representative data in the Supplementary material would be great.
- 3) Supplementary Fig.4, panel d) – do individual dots represent the mean amplitude of Ca²⁺ transients pooled for individual "responder" ROIs or individual astrocytes ?
- 4) Also, it would be good if examples of GCamp5 fluorescence were shown in the Suppl Fig4a. (similar to examples of Fluo-4 images shown in Figs 2,3).
- 5) Supplementary Fig.5 – the quantitative statistical data on correlation between immunostaining with tdTomato and astrocytic and neuronal markers should be shown as diagrams. Does " n = 3" refer to the number of preparations here ? **What were the cell numbers ?**

Reviewer #2 (Remarks to the Author):

This manuscript by Bohmbach et al investigates the role of astrocytes on dendritic integration in the hippocampus. They show that the threshold and amplitude of dendritic spikes are regulated by both D-serine and (endo)cannabinoids. Interestingly, antidromic stimulation of pyramidal neuron's axons in the alveus showed a frequency-dependent regulation of dendritic spikes also mediated by

D-serine and cannabinoids acting on astrocyte CB1 receptors. They propose a signaling mechanism that involves the activation of astrocyte CB1 receptors in response to neuronal activity and D-serine release that ultimately regulate dendritic spikes. Finally, they performed behavioral tests that showed that mice lacking CB1 receptors in astrocytes have deficits in spatial learning. In my opinion this is an interesting and elegant study that uses state-of-the-art methodology and that can have a significant impact in the field. However, I feel that the relationship between astrocyte activity and D-serine is not fully addressed. In addition, the behavioral results lack specificity for hippocampus and they are difficult to link with the electrophysiological results. Therefore, I think this manuscript might be suitable for publication in Nature communications only after major revision.

Major points:

1. For the calcium analysis ROIs were manually selected, the authors need to clearly state what criteria were used to select these ROIs. If I understood correctly, areas that showed a calcium event during stimulation (either for WIN or alveus stimulation) were chosen. However, this approach can bias the analysis because regions that show a spontaneous calcium event during stimulation would be taken as responders. It would be necessary to see if the frequency or amplitude of calcium events in the astrocyte are different before and after stimulation.
2. It is not convincing that the effects of D-serine happen downstream of cannabinoid signaling and astrocyte activation. They show that D-serine occludes the effect of WIN (Fig 2), maybe due to a saturation of NMDA receptors. To show that D-serine is released downstream of cannabinoid signaling they could see if WIN also occludes the effects of D-serine or if WIN effects can be blocked by DAAO inhibition.
3. In addition, to show a relationship between the astrocyte activity and the effects of D-serine, they could to see if D-serine can still regulate dendritic spikes in aCB1ko mice to levels similar to the WT.
4. They show that alveus stimulation at frequencies higher to 20Hz don't have an effect on dendritic spikes and astrocyte calcium, possibly due to a frequency-dependent pyramidal cell excitability conferred by HCN channels. This explanation is convincing for the regulation of dendritic spikes because it is blocked by ZD7288. However, it would be necessary to test whether ZD7288 also blocks astrocyte responses.
5. The authors need to provide an explanation for why they sometimes performed alveus stimulation at 10Hz and other times at 20Hz. In my opinion, pharmacological experiments should be performed always in the same conditions, ideally at 10Hz because it elicits the maximum effect.
6. The behavioral analyses are based on a global astrocyte knock-out mouse line. Therefore, from the data presented, one cannot exclude that the function of astroglial CB1 on spatial memory is not due to actions in other parts of the brain. The authors should perform some experiments in a hippocampal specific knock-out, or try to rescue CB1 specifically in hippocampal astrocytes.
7. Also importantly, the behavioral deficits in aCB1R mice could be due to a signaling mechanism different to the one described in this manuscript. The authors should perform some key experiment to convincingly show that the same processes observed in slices are required for the location memory.
8. The title mentions "spatial leaning". I would suggest using more specific terms like "object or stimulus location". Spatial memory refers more to the memory of the subject's location in space in relationship to a target (eg in the Morris' Water Maze). More importantly, the authors do not show which phase of memory is affected by the mutation. Thus, we do not know if acquisition (learning), consolidation, or recall of memory are touched. The experiments to address point 7, if performed by injecting drugs into mutant animals, could also help answer this point.

Minor

1. It is not clear how the slow component of the dendritic spike was analyzed. From the representative traces they provide it seems that in some cases it can be difficult to differentiate the fast component from the peak of the slow component (for example in Fig 1e)
2. Methods: Animals were kept in 12 h light/dark conditions. However, at what time light was on and at what time animals were sacrificed for electrophysiology or underwent behavioral testing. This is a very important piece of information, because we know that both astrocyte and endocannabinoid activities are under circadian control.

Reviewer #3 (Remarks to the Author):

Using two-photon Ca²⁺ imaging, the authors explored effect of astrocyte activation on the threshold and amplitude of dendritic spikes of hippocampal CA1 pyramidal neurons induced by iontophoretic application of glutamate. They found that antidromic stimulation of CA1 pyramidal neuronal axons, which induced Ca²⁺ elevation in astrocytes, frequency dependently enhanced dendritic spikes, a response blocked by endocannabinoid receptor antagonist, D-serine degradation enzyme, HCN inhibitor, or specific deletion of astrocytic CB1Rs. They further showed that deletion of astrocytic CB1Rs impaired mouse spatial learning. These results disclose a new form of neuron-glia interaction in hippocampal spatial learning through enhancing CA1 dendritic inputs mediated by activity-dependent activation of HCN channels in and release of endocannabinoid from CA1 pyramidal neurons, which in turn activates CB1Rs in and tricks D-serine release from astrocytes that augment NMDA receptor-sensitive dendritic spikes. The experiments are well-designed, the hypothesis is tested by multiple approaches, and evidence provided seems convincing. I only have some minor questions:

1. D-serine is well known involved in the NMDA receptor-dependent synaptic plasticity in CA1 neurons. Did authors examine whether such plasticity contributes to the frequency-dependent augmentation of dendritic spikes reported in this study?
2. What do red and black lines represent respectively in top right panels in Fig 1d and 1i? Similarly , what do blue and black lines represent respectively in top right panels in Fig 1e and 1j?

REPLY TO REVIEWER COMMENTS

Reviewer #1

We were very happy about the positive assessment of our manuscript. We also thank the reviewer for the careful analysis of our manuscript and the helpful comments. We have performed several sets of additional experiments to comprehensively address their points. Please see below for details. Changed text in the manuscript is in red.

[...]

1) The suggested mechanism of glia-driven modulation of dendritic integration is indirect and basal levels of occupancy of NMDAR Glycine-site will affect the dynamic range of this regulatory cascade. To verify that this cascade of glia-neuron interaction occurs via modulation of NMDAR Gly-site, it is important to show that antagonists of this site (such as CGP 78608 or L-701,324) block the effect of astrocytic CB1Rs on dendritic spike threshold and amplitude (as in Figure 2).

We performed the suggested experiments and confirmed the involvement of the NMDAR co-agonist binding site. The new figure panel Fig. 2d shows that blocking the NMDAR co-agonist binding site with the antagonist 5,7-dichlorokynurenic acid (DCKA, 10 μ M) occludes the effect of WIN55 on the dendritic spike threshold and amplitude.

2) The data shown in the Fig.3 and S3, strongly support the role of D-Serine in modulation of dendritic spike firing. These data, however, do not rule out putative release of D-Serine from neurons. Although previous works of the Authors argue very strongly against neurons as a main source of D-Serine, in view of current debate in the field, it is important to substantiate the dominant role for astrocytes. This could be done by demonstrating that intracellular perfusion of astrocytes with Ca-chelators strongly attenuates the effect of alveus stimulation on spike threshold and slow component amplitude.

The revised manuscript includes several new experiments that further support the role of astrocytes in the activity dependent modulation of dendritic integration. We performed two additional sets of experiments using astrocytic expression of a human Ca^{2+} pump (hPMCA2w/b), which is a method for attenuating astrocytic Ca^{2+} signals developed by the Khakh lab (Yu *et al.*, 2018) (CaEx, new Fig. 8, new Suppl. Fig. 9-10). In the new Fig. 8, we now show that this manipulation of astrocytic signaling blocks the activity dependent changes of dendritic integration. Importantly, it also impairs object location memory. Thus, both deletion of astrocytic CB1Rs and astrocytic expression of hPMCA2w/b disrupt the newly discovered positive feedback loop between pyramidal cell activity and their dendritic integration and object location memory. This strengthens the causal relationship between the two and the role of astrocytes. Furthermore, the new figure panels Fig. 4f-g shows that ZD7288 blocks the activity-dependent increase of the astrocytic Ca^{2+} event frequency, which also implies that astrocytic Ca^{2+} signals are part of the negative feedback loop.

In addition, we would like to point out that we did not and do not claim that astrocytes have a dominant role in release of D-serine, because we have not directly compared astrocytes to other potential sources of D-serine. (This is indeed an interesting topic for another study.) Our aim was to test if activity-dependent release of D-serine from astrocytes modifies dendritic integration of synaptic input and behaviors, for which dendritic spiking of CA1 pyramidal cells is thought be

important. Nonetheless, we maintain that there is overwhelming evidence from many labs and brain regions that astrocytes are a key regulator of extracellular D-serine levels (Yang *et al.*, 2003; Panatier *et al.*, 2006; Henneberger *et al.*, 2010; Takata *et al.*, 2011; Rasooli-Nejad *et al.*, 2014; Kronschläger *et al.*, 2016; Papouin *et al.*, 2017a; Adamsky *et al.*, 2018; Koh *et al.*, 2021) even though much is to be learnt about the cellular routes and sources of D-serine supply and metabolisms and the relevant release mechanisms (Wolosker *et al.*, 2016; Papouin *et al.*, 2017b). We have modified the discussion and updated relevant references to make the latter that clearer.

3) Theoretically, there are several alternative mechanisms by which astrocytes could influence neuronal excitability and dendritic integration. Astrocytes can affect the GABAergic inhibition by releasing GABA [Lee, Yoon *et al.* Science 2010] or via release of ATP causing down-regulation of GABAA receptors [Lalo *et al.* PloSBiology 2014]. Astrocytes were also shown to synchronize neuronal firing [Pascual 2005] and affect trafficking of postsynaptic AMPA receptors by release of ATP/adenosine [Pougnat 2014 Neuron, Rodrigues 2015]. Could Authors rule out the putative involvement of these mechanisms in astroglial control of dendritic integration?

All experiments in acute slices were performed in the presence of the GABAA receptor antagonist picrotoxin so their involvement can be excluded in these experiments. This was only briefly mentioned in the methods section. We have added another statement at the beginning of the results section. A role of short-term AMPA receptor surface expression appears unlikely, because we did not observe changes of the mini-like AMPA EPSPs in uncaging and iontophoretic experiments (see results page 6, new added statement on page 10).

That does not exclude that other signaling cascades downstream of astrocytic Ca²⁺ signals play a role in other experimental settings. We now point this out at the end of the discussion.

4) The intracellular Ca²⁺-elevation in astrocytes is instrumental for the glia-neuron interactions reported in the paper. [...] Still, many details of Ca²⁺-measurements need clarification to allow wider audience fully appreciate the importance of CB1R-mediated astroglial signalling:

(i) Although the Methods mention evaluation of the OVERALL changes in astrocytic Ca²⁺ signals (lines 757, 765), further description and illustrations suggest that Ca²⁺-signalling was evaluated in just one (Figs 2 and 3) or few (Fig.S4) ROIs per cell. Do the values of R/RO ratio reported in Figs. 2b and 3c refer to the whole astrocytes or just selected “responder” ROIs? The similar question is applicable to the frequency of Ca²⁺-transients reported in the Fig.S4.

Our previous description of the experiments was indeed lacking necessary details. We used several different approaches for imaging astrocytic Ca²⁺ signaling and for its analysis throughout the manuscript. In several experiments, we analyzed a single large region of interest (ROI) covering the entire astrocyte but sparing the soma and major branches for each cell. This was done to avoid defining complex criteria for event detection or ROI placement and thus to avoid these potential biases. Amplitudes are calculated over the entire ROI. We have revised the manuscript in several places to improve clarity.

For data depicted in Fig. 2a-b, a ROI covering the entire astrocyte but excluding its soma and major branches were analyzed for each astrocytes.

For Fig. 3a-c, a ROI covering the entire astrocyte but excluding the soma and major branches was analyzed for each astrocyte. A total of 13 recordings were performed. 9 of those cells were

responding. For these responders, the effect of AM251 was quantified. This is now clearly stated in the figure legend.

For Fig. 4a-c, a ROI covering entire astrocytes but excluding the soma and major branches was analyzed for each astrocyte. We now clearly state the number of analyzed cells and responders.

For Suppl. Fig. 4a-d, we used manual identification of Ca²⁺ transients and quantified their frequencies and amplitudes in each section of the recording per cell. Because of its greater subjectivity, we placed this analysis in the supplementary material.

For the new Fig. 4f-g and Suppl. Fig. 5, AQuA (Wang *et al.*, 2019) was used.

(ii) To verify that reported data represent the general pattern of changes in astrocytic signalling, Authors need to specify how ROIs were defined and selected for analysis, how many ROIs were analysed per single astrocytes and how the average number of “responders” per cell changes upon stimulation of alveus and application of CB1R modulators;

This question is immediately related to the previous one and our answer to that applies here too. Because we used a ROI covering the entire astrocyte but not the soma and major branches in the relevant recordings, a cell can be a responder or not in a section of recording (single ROI per astrocyte). In Fig. 3c, the number and percentages of responders are now given in the figure and/or legend. The pharmacological analysis was only performed on cells that responded to the stimulation. In Fig. 4b, astrocytes were analyzed using a single ROI. The percentage and numbers of astrocytes responding to 10 or 40 Hz stimulation is now given in the figure and legend, respectively. Their average response Fig. 4c, is calculated over all astrocytes. This is now obvious from the legend.

Throughout the manuscript, an astrocyte was defined as ‘responding’ when the relative change in fluorescence intensity was higher than 5 % compared to baseline levels. This is stated several times throughout the manuscript.

(iii) Regarding the signalling microdomains, has any kind of spatial analysis of local elevations in fluorescence (e.g. particle analysis) been performed prior to selection of ROIs ?

This is an interesting point. We had not performed such an analysis for the initial manuscript. The revised manuscript contains a completely new data set (Fig. 4f-g, Suppl. Fig. 5), which was analyzed using AQuA (Wang *et al.*, 2019). The results from the control recordings with 10 Hz alveus stimulation fully support our original conclusions in the previous version of the manuscript.

(iv) Overall magnitude of Ca²⁺-elevation: Average amplitude of Ca-transients elicited in astrocytes by alveus stimulation and measured with Fluo-4 (about 25-30% of baseline, Fig.3) appears to be considerable smaller than of transients assessed with genetically-encoded GCamp5d (about 50-70% of baseline, Fig.S4), which might be explained either by buffering of Ca²⁺ with high concentration of Fluo-4 or by preferential expression of GCamp in the vicinity of cell membrane (or both). In Authors opinion, which approach provides the better estimate of cytosolic Ca²⁺-elevation ? Also, do Authors believe that elevation of 30-50% from the baseline Ca²⁺-level is enough to activate a significant release of D-Serine?

This is an excellent question. In our manuscript, we have used ‘qualitative’ Ca²⁺ imaging to test if astrocytic Ca²⁺ signals change overall upon stimulation and in certain conditions. This is sufficient for the conclusions we wanted to draw. Comparing amplitudes between these individual experiments is difficult because of varying indicator properties and concentrations.

For obtaining a quantitative relationship between a local astrocytic Ca²⁺ transients and a released molecular such as D-serine, both need to be observed simultaneously. We have previously established the toolset for quantitative Ca²⁺ imaging in astrocytes (King *et al.*, 2020). We have also recently contributed to the development of a first optical sensor for D-serine (Vongsouthi *et al.*, 2021). However, this sensor is a first prototype and not yet sensitive enough for this type of experiment. We are currently optimizing it, so we will hopefully be able to do this challenging experiment in the future. We put a note at the end of the discussion that the relationship between astrocytic Ca²⁺ concentration changes and gliotransmitter release remains to be fully established.

5) Spatial specificity of the reported feedback mechanism is a bit puzzling and needs further discussion: The release of endocannabinoids from CA1 neurons onto stratum radiatum astrocytes was caused by stimulation of alveus. Would stimulation of the Schaffer collaterals result in similar Ca²⁺-elevation and release of D-Serine?

This is an interesting point. Simply put, our experiments using iontophoretic stimulation of synaptic input cannot be combined with stimulation of synaptic input in a useful manner. However, our previous study demonstrated that a high-frequency stimulation of Schaffer collaterals increases NMDAR co-agonist levels in an astrocyte-dependent manner, also involving astrocytic Ca²⁺ signaling (Henneberger *et al.*, 2010). So, the direct answer to the question is yes; we had cited our previous work.

However, the question about the spatial arrangement cannot be answered currently as far as we are aware of. There are several ‘spatial’ variables that we do not know the value of: for example, the density and degree of dendritic depolarization, the range of action of released endocannabinoids, the range of action of released D-serine. Our working hypothesis is that the distributed activity of a significant number of pyramidal cells leads to distributed dendritic depolarizations, which in turn triggers distributed release of endocannabinoids, D-serine increases, and increases of dendritic spiking very broadly in the tissue. This is a likely scenario, because we have very little control over the spatial arrangement of the probed dendrites and active astrocytes and the effect is robust (no clear failure/success distribution, see Fig. 3d-e). We have added a statement to the discussion alluding to the open questions about spatial extent of astrocytic Ca²⁺ transients and the range of action of released gliotransmitter.

Minor/technical issues:

1) technical issues: (i) at what wavelengths the Fluo-4 and Alexa594 signals were detected? - to ensure the putative cross-bleed between red and green channels did not affect the “responder” detection.

Two sets of filters were used (Scientifica microscope: 565 nm dichroic, 500-550 nm, 590-650 nm; Olympus microscope: 570 nm dichroic, 515-560 nm, 575-630 nm). This is now stated in the methods section. We found no indication of relevant bleed-through between channels in our current study or previous studies using the same configuration, for instance (Minge *et al.*, 2017, 2021; King *et al.*, 2020).

2) In the whole-cell experiments in astrocytes, their identity was conferred by the presence of the gap-junction coupling. How this was assessed, via spreading of fluorescent dye or by paired electrophysiological recordings? Showing some representative data in the Supplementary material would be great.

This was indeed assessed by dye spreading from the patched astrocyte into neighboring astrocytes. We added the new Suppl. Fig. 11 to illustrate the typical current and voltage responses and an image of a gap junction coupled network.

3) Supplementary Fig.4, panel d) – do individual dots represent the mean amplitude of Ca²⁺ transients pooled for individual “responder” ROIs or individual astrocytes ?

Individual data points represent the mean Ca²⁺ transient amplitude in an astrocyte (Suppl. Fig. 4e, previously Suppl. Fig. 4d). This is now stated in the figure legend.

4) Also, it would be good if examples of GCamp5 fluorescence were shown in the Suppl Fig4a. (similar to examples of Fluo-4 images shown in Figs 2,3).

We have added a representative example (new panel Suppl. Fig. 4c).

5) Supplementary Fig.5 – the quantitative statistical data on correlation between immunostaining with tdTomato and astrocytic and neuronal markers should be shown as diagrams. Does “ n = 3” refer to the number of preparations here ? What were the cell numbers ?

“n = 3” refers to the number of animals. Per animal three 512 x 512 μm sections of 20 μm depth were analyzed and averaged. We have added this information and further analyses to the figure legend.

Reviewer #2 (Remarks to the Author):

Thank for the constructive comments and the positive evaluation. We have performed several sets of new experiments and fully addressed your points as described below. All changes made to the manuscript are in red.

[...]

1. For the calcium analysis ROIs were manually selected, the authors need to clearly state what criteria were used to select these ROIs. If I understood correctly, areas that showed a calcium event during stimulation (either for WIN or alveus stimulation) were chosen. However, this approach can bias the analysis because regions that show a spontaneous calcium event during stimulation would be taken as responders. It would be necessary to see if the frequency or amplitude of calcium events in the astrocyte are different before and after stimulation.

We have carefully revised the description of the Ca²⁺ imaging experiments and analyses throughout the manuscript to improve clarity. We used several strategies for experiments and analyses. In several main experiments (Fig. 2a-b, Fig. 3a-c, Fig. 4a-c), we used a single ROI per astrocyte sparing major branches and the cell body (example in Fig. 4a). We did so because we wanted to avoid potential biases introduced by complex ROI definitions or event detection settings and because we were interested whether the overall responsiveness was affected by our manipulations. This is more clearly stated now.

Responders were those cells, in which such a global signal changed > 5% as stated (e.g. Fig3c, left panel). In Fig. 3c (right panel), the effect of pharmacological manipulations was only analyzed in those responders.

Because this approach likely underestimates the amplitude of individual Ca²⁺ transients, we had also provided an analysis using manual analysis of Ca²⁺ imaging data (Suppl. Fig. 4). The results were qualitatively the same.

In the revised manuscript, we now provide a new dataset of astrocytic Ca²⁺ that was analyzed using automated detection of Ca²⁺ events (AQuA) (Wang *et al.*, 2019). We reproduced our previous finding that 10 Hz stimulation increases the frequency of Ca²⁺ transients (control recordings in the new Fig. 4f-g) (and also their amplitude, Suppl. Fig. 5).

2. It is not convincing that the effects of D-serine happen downstream of cannabinoid signaling and astrocyte activation. They show that D-serine occludes the effect of WIN (Fig 2), maybe due to a saturation of NMDA receptors. To show that D-serine is released downstream of cannabinoid signaling they could see if WIN also occludes the effects of D-serine or if WIN effects can be blocked by DAAO inhibition.

We performed the suggested experiment. The new figure panel 2f shows that the WIN effect is blocked by DAAO. In addition, the WIN effect is also prevented by blocking the NMDAR co-agonist binding site with DCKA (new figure panel 2d). Both experiments indicate that the WIN effect on dendritic integration involves the NMDAR co-agonist binding site.

3. In addition, to show a relationship between the astrocyte activity and the effects of D-serine, they could to see if D-serine can still regulate dendritic spikes in aCB1ko mice to levels similar to the WT.

As suggested, we applied saturating concentrations of D-serine to acute hippocampal slices of aCB1KO mice while investigating dendritic spiking. The results have been added to the revised Fig. 5 (panels c and d, red dots and bars). They show that the effect of D-serine in aCB1KO mice is indeed comparable to the stimulation of the alveus in WT mice.

4. They show that alveus stimulation at frequencies higher to 20Hz don't have an effect on dendritic spikes and astrocyte calcium, possibly due to a frequency-dependent pyramidal cell excitability conferred by HCN channels. This explanation is convincing for the regulation of dendritic spikes because it is blocked by ZD7288. However, it would be necessary to test whether ZD7288 also blocks astrocyte responses.

We performed the suggested experiments using an automated approach for the analysis of the astrocytic calcium imaging data. Using viral expression GCaMP6f we reproduced our findings, that 10

Hz stimulation of the alveus leads to an increased calcium event frequency (and also amplitude in this case). Further, we now show that ZD7288 blocked the observed effects (see new figure panels 4f-g and the new Suppl. Fig. 5).

5. The authors need to provide an explanation for why they sometimes performed alveus stimulation at 10Hz and other times at 20Hz. In my opinion, pharmacological experiments should be performed always in the same conditions, ideally at 10Hz because it elicits the maximum effect.

This has historical reasons. We simply tested 20 Hz first and later moved on to testing several other frequencies. However, many key observations have been performed at 10 Hz alveus stimulation: effect on dendritic spiking (e.g. Fig. 4d-e), CB1R dependence (Fig. 4d-e, Fig. 5c-d), D-serine involvement (Fig. 5c-d), role of Ca²⁺ signaling (Fig. 4a-c, Fig. 4f-g, Fig. 8b-c). It is also worth noting that the characterization of the basic cellular phenomenon, i.e., the CB1R-dependent increase of D-serine levels promoting dendritic spiking was first characterized without alveus stimulation (Fig. 1-2).

6. The behavioral analyses are based on a global astrocyte knock-out mouse line. Therefore, from the data presented, one cannot exclude that the function of astroglial CB1 on spatial memory is not due to actions in other parts of the brain. The authors should perform some experiments in a hippocampal specific knock-out, or try to rescue CB1 specifically in hippocampal astrocytes.

This is an important and often neglected point, although it has been shown previously that object location memory critically depends on the hippocampus (Assini *et al.*, 2009; Barker & Warburton, 2011) which is why we chose this test. As suggested, we have tried several published approaches based on AAVs for local expression of cre in astrocytes. So far, we have not been able to find a protocol that reliably leads to expression of a recombination reporter in astrocytes in a large part of the hippocampus but without reporter expression also in neurons, especially near the AAV injection site. As an alternative, we used an approach developed by the Khakh lab, in which expression of a human Ca²⁺ pump in astrocytes attenuates astrocytic Ca²⁺ signaling (CalEx) (Yu *et al.*, 2018). We successfully established the technique and demonstrate in the revised manuscript that this manipulation of hippocampal astrocytes disrupts the activity-dependent modulation of dendritic spiking and object location memory. These experiments demonstrate that interfering with astrocytic Ca²⁺ signaling disrupts the feedback loop between pyramidal cell activity and their dendritic integration, which as a link between astrocytic Ca²⁺ signals and NMDAR co-agonist levels has not been demonstrated in this manuscript, but see for instance (Henneberger *et al.*, 2010; Rasooli-Nejad *et al.*, 2014; Papouin *et al.*, 2017a; Robin *et al.*, 2018). It is also the requested local manipulation of hippocampal astrocytes that disrupts object location memory. The results are presented in the new Fig. 8 and new Suppl. Fig. 9-10.

7. Also importantly, the behavioral deficits in aCB1R mice could be due to a signaling mechanism different to the one described in this manuscript. The authors should perform some key experiment to convincingly show that the same processes observed in slices are required for the location memory.

In the experiments explained above (new Fig. 8) we provide a second line of evidence that disruption of an astrocyte-dependent increase of dendritic spiking is important for object location memory. We have also included a statement in the discussion that other signaling pathways can be activated downstream of astrocytic CB1R and Ca²⁺ and that this needs consideration in future studies.

We are sure that the reviewer is not expecting us to test experimentally and in vivo the involvement of the host of alternative signaling pathways that can be downstream of astrocytic CB1Rs and Ca²⁺ signaling. We have recently reviewed the challenges that lie ahead for this research field (Bohmbach *et al.*, 2022).

8. The title mentions "spatial leaning". I would suggest using more specific terms like "object or stimulus location". Spatial memory refers more to the memory of the subject's location in space in relationship to a target (eg in the Morris' Water Maze). More importantly, the authors do not show which phase of memory is affected by the mutation. Thus, we do not know if acquisition (learning), consolidation, or recall of memory are touched. The experiments to address point 7, if performed by injecting drugs into mutant animals, could also help answer this point.

We did not fully understand this comment. The formation of spatial memory, i.e., spatial learning is needed for the formation of a cognitive map of the environment or test arena and for recording the location of objects and landmarks within it. This is true for the object location arena (objects) and for the Morris Water Maze (platform), except that the latter involves an aversive stimulus (water) which is however not a conceptual difference. In both cases, allocentric spatial orientation plays a dominant role, less so the egocentric component or relative position. More important maybe: Our behavioral test in the O-maze reveals a deficit of aCB1KO mice to update the location of an aversive stimulus (air puff). In both tests we used, spatial memory is required. Therefore, we believe our current manuscript title (spatial learning) is adequate. However, we feel that this is a relatively minor point and are happy to discuss this further.

We are aware of the complexities of memory processes (Bohmbach *et al.*, 2022). That is why we made and make no claims regarding the link between the observed cellular and network phenomena and specific memory processes (such as acquisition, retention, recall).

Minor

1. It is not clear how the slow component of the dendritic spike was analyzed. From the representative traces they provide it seems that in some cases it can be difficult to differentiate the fast component from the peak of the slow component (for example in Fig 1e)

The fast component is easily identifiable in the first derivative (see for instance Fig. 1). It was used for orientation to accurately set the windows in which both were analyzed. This is now stated in the results section.

2. Methods: Animals were kept in 12 h light/dark conditions. However, at what time light was on and at what time animals were sacrificed for electrophysiology or underwent behavioral testing. This is a very important piece of information, because we know that both astrocyte and endocannabinoid activities are under circadian control.

All acute slices were prepared in the late morning, which corresponds to the beginning of the non-active light phase of the animals. This is now stated in the methods section. Behavioral experiments were performed at the beginning of the dark phase with mice living under reversed 12h light/dark conditions. This had been stated.

We are aware of the literature on how sleep, arousal and circadian rhythms control various aspects of astrocyte physiology (among many others Papouin *et al.*, 2017a; Bojarskaite *et al.*, 2020; McCauley *et al.*, 2020). It will certainly be interesting to test such interdependencies in the future.

Reviewer #3 (Remarks to the Author):

We thank the reviewer for the positive and helpful comments on our manuscript. We fully address their minor comments in the revised manuscript. Changes in the text are marked in red.

[...]

1. D-serine is well known involved in the NMDA receptor-dependent synaptic plasticity in CA1 neurons. Did authors examine whether such plasticity contributes to the frequency-dependent augmentation of dendritic spikes reported in this study?

The effect of alveus stimulation on dendritic spiking was temporary (return to baseline after 5 minutes, Suppl. Fig. 3). We also did not observe a change of miniature-like AMPAR-mediated EPSPs after D-serine application and alveus stimulation (results, pages 6 and added statement on page 10). Therefore, these manipulations do not induce synaptic plasticity on their own. However, in the presence of ongoing synaptic activity (e.g., in vivo) increased D-serine release would promote dendritic spiking and synaptic plasticity.

2. What do red and black lines represent respectively in top right panels in Fig 1d and 1i? Similarly, what do blue and black lines represent respectively in top right panels in Fig 1e and 1j?

Thank you for the comment. This was indeed not explained in the figure legend. In d, e, i and j the upper right panels illustrate traces recorded with the *lowest stimulus eliciting a dendritic spike under baseline conditions* (black trace before drug, colored trace in the presence of drug). The lower right panels illustrate traces recorded with the *lowest stimulus eliciting a dendritic spike in the presence of the drug* (black trace before drug, colored trace in the presence of drug). For instance, the upper right panel in Fig. 1d shows a dendritic spike elicited with the threshold stimulus under control conditions (black trace) and the response obtained in APV (red traces, no dendritic spike with that stimulus strength).

We added this explanation to the legend of Figure 1.

References

- Adamsky A, Kol A, Kreisel T, Doron A, Ozeri-Engelhard N, Melcer T, Refaeli R, Horn H, Regev L, Groysman M, London M & Goshen I (2018). Astrocytic Activation Generates De Novo Neuronal Potentiation and Memory Enhancement. *Cell* **174**, 59–71.
- Assini FL, Duzzioni M & Takahashi RN (2009). Object location memory in mice: Pharmacological validation and further evidence of hippocampal CA1 participation. *Behav Brain Res* **204**, 206–211.
- Barker GRI & Warburton EC (2011). When Is the Hippocampus Involved in Recognition Memory? *J Neurosci* **31**, 10721–10731.
- Bohmbach K, Henneberger C & Hirrlinger J (2022). Astrocytes in memory formation and maintenance. *Essays Biochem* EBC20220091.
- Bojarskaite L, Bjørnstad DM, Pettersen KH, Cunen C, Hermansen GH, Åbjørsbråten KS, Chambers AR, Sprengel R, Vervaeke K, Tang W, Enger R & Nagelhus EA (2020). Astrocytic Ca²⁺ signaling is reduced during sleep and is involved in the regulation of slow wave sleep. *Nat Commun* **11**, 3240.
- Henneberger C, Papouin T, Oliet SHR & Rusakov DA (2010). Long-term potentiation depends on release of d-serine from astrocytes. *Nature* **463**, 232–236.
- King CM, Bohmbach K, Minge D, Delekate A, Zheng K, Reynolds J, Rakers C, Zeug A, Petzold GC, Rusakov DA & Henneberger C (2020). Local Resting Ca²⁺ Controls the Scale of Astroglial Ca²⁺ Signals. *Cell Rep* **30**, 3466–3477.
- Koh W et al. (2021). Astrocytes Render Memory Flexible by Releasing D-Serine and Regulating NMDA Receptor Tone in the Hippocampus. *Biological Psychiatry*; DOI: 10.1016/j.biopsych.2021.10.012.
- Kronschlager MT, Drdla-Schutting R, Gassner M, Honsek SD, Teuchmann HL & Sandkuhler J (2016). Gliogenic LTP spreads widely in nociceptive pathways. *Science* **354**, 1144–1148.
- McCauley JP, Petroccione MA, D’Brant LY, Todd GC, Affinnih N, Wisnoski JJ, Zahid S, Shree S, Sousa AA, De Guzman RM, Migliore R, Brazhe A, Leapman RD, Khmaladze A, Semyanov A, Zuloaga DG, Migliore M & Scimemi A (2020). Circadian Modulation of Neurons and Astrocytes Controls Synaptic Plasticity in Hippocampal Area CA1. *Cell Rep* **33**, 108255.
- Minge D, Domingos C, Unichenko P, Behringer C, Pauletti A, Anders S, Herde MK, Delekate A, Gulakova P, Schoch S, Petzold GC & Henneberger C (2021). Heterogeneity and Development of Fine Astrocyte Morphology Captured by Diffraction-Limited Microscopy. *Front Cell Neurosci* **15**, 669280.
- Minge D, Senkov O, Kaushik R, Herde MK, Tikhobrazova O, Wulff AB, Mironov A, van Kuppevelt TH, Oosterhof A, Kochlamazashvili G, Dityatev A & Henneberger C (2017). Heparan Sulfates Support Pyramidal Cell Excitability, Synaptic Plasticity, and Context Discrimination. *Cereb Cortex* **27**, 903–918.
- Panatier A, Theodosis DT, Mothet J-P, Touquet B, Pollegioni L, Poulain DA & Oliet SHR (2006). Glia-derived D-serine controls NMDA receptor activity and synaptic memory. *Cell* **125**, 775–784.

- Papouin T, Dunphy JM, Tolman M, Dineley KT & Haydon PG (2017a). Septal Cholinergic Neuromodulation Tunes the Astrocyte-Dependent Gating of Hippocampal NMDA Receptors to Wakefulness. *Neuron* **94**, 840–854.
- Papouin T, Henneberger C, Rusakov DA & Oliet SHR (2017b). Astroglial versus Neuronal D-Serine: Fact Checking. *Trends Neurosci* **40**, 517–520.
- Rasooli-Nejad S, Palygin O, Lalo U & Pankratov Y (2014). Cannabinoid receptors contribute to astroglial Ca²⁺-signalling and control of synaptic plasticity in the neocortex. *Phil Trans R Soc B* **369**, 20140077.
- Robin LM et al. (2018). Astroglial CB1 Receptors Determine Synaptic D-Serine Availability to Enable Recognition Memory. *Neuron* **98**, 935–944.
- Takata N, Mishima T, Hisatsune C, Nagai T, Ebisui E, Mikoshiba K & Hirase H (2011). Astrocyte calcium signaling transforms cholinergic modulation to cortical plasticity in vivo. *J Neurosci* **31**, 18155–18165.
- Vongsouthi V, Whitfield JH, Unichenko P, Mitchell JA, Breithausen B, Khersonsky O, Kremers L, Janovjak H, Monai H, Hirase H, Fleishman SJ, Henneberger C & Jackson CJ (2021). A Rationally and Computationally Designed Fluorescent Biosensor for d-Serine. *ACS Sens*; DOI: 10.1021/acssensors.1c01803.
- Wang Y, DelRosso NV, Vaidyanathan TV, Cahill MK, Reitman ME, Pittolo S, Mi X, Yu G & Poskanzer KE (2019). Accurate quantification of astrocyte and neurotransmitter fluorescence dynamics for single-cell and population-level physiology. *Nat Neurosci* **22**, 1936–1944.
- Wolosker H, Balu DT & Coyle JT (2016). The Rise and Fall of the d-Serine-Mediated Gliotransmission Hypothesis. *Trends Neurosci* **39**, 712–721.
- Yang Y, Ge W, Chen Y, Zhang Z, Shen W, Wu C, Poo M & Duan S (2003). Contribution of astrocytes to hippocampal long-term potentiation through release of D-serine. *Proc Natl Acad Sci USA* **100**, 15194–15199.
- Yu X, Taylor AMW, Nagai J, Golshani P, Evans CJ, Coppola G & Khakh BS (2018). Reducing Astrocyte Calcium Signaling In Vivo Alters Striatal Microcircuits and Causes Repetitive Behavior. *Neuron* **99**, 1170–1187.

REVIEWERS' COMMENTS

Reviewer #1 (Remarks to the Author):

The manuscript has been considerably improved after revision. All reviewers' queries have been carefully addressed. Authors extended their experimental evidence to address the reviewers' comments, the data presented fully support the conclusions and strengthen the overall message of the paper.

Reviewer #3 (Remarks to the Author):

The revised manuscript has addressed all my comments. I agree that the manuscript is now ready to be published in Nature Communication.

Point-by-point response

Editorial recommendation

... Based on their comments, editorially we recommend to expand your discussion to put your results in the context of previous data on impact of CB1 receptors and astrocytes on acquisition and consolidation of memory (e.g. Busquets-Garcia 2016, PNAS, 113:9904-9909). ...

The suggested paper is not about astrocytic CB1 receptors, or a behaviour related to those we studied. However, the suggested author has recently published two other relevant papers, which we now include in our discussion.

'They also add to the growing experimental evidence indicating that astrocytic CB1Rs are an important regulator of a variety of complex behaviors¹⁻³.'

1. Jimenez-Blasco, D. *et al.* Glucose metabolism links astroglial mitochondria to cannabinoid effects. *Nature* **583**, 603–608 (2020).
2. Robin, L. M. *et al.* Astroglial CB1 Receptors Determine Synaptic D-Serine Availability to Enable Recognition Memory. *Neuron* **98**, 935–944 (2018).
3. Ramon-Duaso, C., Conde-Moro, A. R. & Busquets-Garcia, A. Astroglial cannabinoid signaling and behavior. *Glia* **71**, 60–70 (2023).

Reviewer #1 (Remarks to the Author)

The manuscript has been considerably improved after revision. All reviewers' queries have been carefully addressed. Authors extended their experimental evidence to address the reviewers' comments, the data presented fully support the conclusions and strengthen the overall message of the paper.

We were delighted to read the very positive assessment of our revision.

Reviewer #3 (Remarks to the Author)

The revised manuscript has addressed all my comments. I agree that the manuscript is now ready to be published in Nature Communication.

We were happy to read that the reviewer recommends publication of our manuscript.